

# Modelling chemical advection during magma ascent

Hugo Dominguez [1], Nicolas Riel [2], and Pierre Lanari [1]

[1]Institute of Geological Sciences, University of Bern, Baltzestrasse 1+3, CH-3012 Bern, Switzerland
[2]Institute of Geosciences, Johannes Gutenberg-University, Mainz, Germany

**Correspondence:** Hugo Dominguez (hugo.dominguez@unibe.ch)

**Abstract.**

Modelling magma transport requires robust numerical schemes for chemical advection. Current numerical schemes vary in their ability to be mass conservative, computationally efficient, and accurate. This study compares four of the most commonly used numerical schemes for advection: an upwind scheme, a weighted essentially non-oscillatory (WENO-5) scheme, a semi-Lagrangian (SL) scheme, and a marker-in-cell (MIC) method. We assess the behaviour of these schemes using the passive advection of two different magmatic compositions. This is coupled in 2D with the temporal evolution of a melt anomaly that generates porosity waves. All algorithms, except the upwind scheme, are able to predict the melt composition with reasonable accuracy. In terms of total running time, the upwind and SL schemes are the fastest, and the MIC scheme is the slowest. The WENO-5 scheme shows intermediate total running time but has the lowest amount of mass loss and therefore is best suited for this problem.

## 1 Introduction

Mechanisms of magma ascent and emplacement within the lithosphere and upper asthenosphere remain largely unconstrained (e.g., Connolly and Podladchikov, 2007b; Katz et al., 2022). Studies have attempted to address this problem using techniques ranging from geophysical measurements of the present-day lithosphere to geochemical analysis of the rock record . However, geophysical studies are hampered by indirect measurements, and natural samples in geochemical studies represent only the end-product of the melting processes (Brown, 2013; Clemens et al., 2022; Johnson et al., 2021). Comparatively, numerical modelling allows investigation of these processes at a range of scales in space and time (e.g., Keller, 2013; Katz and Weatherley, 2012).

To numerically model such open systems, it is necessary to be able to describe the chemical and physical processes responsible for magma ascent in a rock. At low melt fractions and in the absence of externally applied stress, the physical processes are based on the continuum formulation of two-phase flow. It takes into account the concurrent mechanisms of rock matrix compaction and buoyancy of partial melt in an interconnected porous network (e.g., Scott and Stevenson, 1984; McKenzie, 1984). This formulation is based on mass and momentum conservation, and an appropriate set of constitutive relationships. In addition, conservation of energy needs to be ensured to link mechanical to chemical processes (e.g., Katz, 2008). Chemical processes, such as phase reactions, can be considered using thermodynamics and/or kinetics and relate the equilibration of the melt with the hosting rock (e.g., Omlin et al., 2017; Bessat et al., 2022). They contribute to the transport dynamics by changing





rock properties, such as density, viscosity, porosity and permeability (Jha et al., 1994; Aharonov et al., 1995b; Keller and Katz, 2016). However, the amount of melt interacting with the rock is also modulated by transport mechanisms (Kelemen et al., 1997; Spiegelman and Kenyon, 1992; Aharonov et al., 1995a). Therefore, the development of integrated models that success-

30 fully describe the complex interaction between reaction and transport is key to understanding melting and melt extraction at all scales.

Numerous numerical studies have investigated reactive melt transport. It has been shown that melts that partially crystallise or dissolve the host rock could be a viable mechanism for channelling flow and creating heterogeneities in the mantle, in the context of oceanic middle ridges (Aharonov et al., 1997; Spiegelman et al., 2001) and sub-arc mantle (Bouilhol et al., 2011).

Concerning lower crust melting, this approach has mainly been used to understand the processes of chemical differentiation and the compositional range of magma in mafic systems (e.g., Jackson et al., 2005; Solano et al., 2012; Riel et al., 2019).

One challenge of reactive melt transport modelling is the advection of the melt composition through its ascent. This part, which is mathematically trivial as it is described by a simple mass balance equation, is numerically challenging (e.g., LeVeque, 1992). This is mainly due to the fact that most numerical models are based on an Eulerian frame of reference, where the

40 discretised space is fixed in space and in time. In contrast, transport is by essence better defined from a Lagrangian perspective, where the observer follows the particles of fluid as they move. In addition, two-phase flow models are at least 2D problems due to the formation of channels (e.g., Barcilon and Lovera, 1989; Connolly and Podladchikov, 2007b) and to the fact that mass cannot be transported efficiently in 1D in the melt (Jordan et al., 2018). This brings limitation to the resolution of the models and hence requires accurate advection schemes.

This study compares four numerical schemes applied to the problem of the advection of magmatic composition: an upwind scheme, a weighted essentially non-oscillatory (WENO) scheme, a semi-Lagrangian (SL) scheme and a marker-in-cell (MIC) method. This selection provides a good representation of the different approaches to solving advection problems that are commonly used in a wide range of applications. We assess the performance of each scheme in terms of accuracy, mass conservation and computational time. A 2D model that couples chemical advection with a two-phase flow solver is then used to evaluate

which algorithm is best suited to this problem.

## 2 Governing equations

Chemical transport in two-phase flow systems is described by the four mass conservation equations of the system (e.g., Aharonov et al., 1997). The first two equations describe the conservation of the total mass of the solid and the liquid:

$$\frac{\partial \left[ (1-\phi)\rho_{\mathrm{s}} \right]}{\partial t} = -\nabla \cdot \left[ (1-\phi)\boldsymbol{v_{\mathrm{s}}}\rho_{\mathrm{s}} \right], \tag{1}$$

$$\frac{\partial (\phi\rho_{\mathrm{f}})}{\partial t} = -\nabla \cdot (\phi\boldsymbol{v_{\mathrm{f}}}\rho_{\mathrm{f}}), \tag{2}$$





where f and s are the fluid and solid phases, $t$ is the time (in s), $\phi$ is the fluid-filled porosity, $\rho$, the density of the respective phase (kg.s$^{-3}$), $v$, the velocity of the respective phase (m.s$^{-1}$). The last two equations express the conservation of each chemical component within the solid and fluid phases:

$$\frac{\partial\left[(1-\phi)\rho_{\mathrm{s}}C_e^{\mathrm{s}}\right]}{\partial t} = \nabla \cdot \left[D_e^{\mathrm{s}}(1-\phi)\rho_{\mathrm{s}}\nabla C_e^{\mathrm{s}} - (1-\phi)\boldsymbol{v_{\mathrm{s}}}\rho_{\mathrm{s}}C_e^{\mathrm{s}}\right], \tag{3}$$

$$\frac{\partial\left(\phi\rho_{\mathrm{f}}C_e^{\mathrm{f}}\right)}{\partial t} = \nabla \cdot \left(\mathbf{D_e^f}\phi\rho_{\mathrm{f}}\nabla C_e^{\mathrm{f}} - \phi\boldsymbol{v_{\mathrm{f}}}\rho_{\mathrm{f}}C_e^{\mathrm{f}}\right), \tag{4}$$

where $C_e$ is the mass fraction of the component $e$ in the respective phase, $D_e^{\mathrm{s}}$ is the solid diffusion coefficient of the component $e$ (in m$^2$.s$^{-1}$) and $\mathbf{D_e^f}$, the hydrodynamic dispersion tensor of the component $e$ in the fluid (m$^2$.s$^{-1}$). These four equations assume no mass transfer due to reactions between the solid and the liquid phases.

## 2.1 Simplifications

In this study, the advection of the chemical components transported by the liquid phase is considered and the diffusion term in Eqs. (3) and (4) is neglected. Since $\rho_{\mathrm{s}}$ is assumed to be constant and that the host rock has a fixed composition, Eq. (3) is omitted.

Subtracting Eq. (2) in Eq. (4), and dividing by $\rho_{\mathrm{f}}$ and $\phi$ yields:

$$\frac{\partial(C_e^{\mathrm{f}})}{\partial t} = -\boldsymbol{v_{\mathrm{f}}}\nabla \cdot (C_e^{\mathrm{f}}). \tag{5}$$

Equation (5) is formally equivalent to Eq. (4) without the dispersion term. Moreover, Eq. (4) is written in conservative form, whereas Eq. (5) is expressed in Lagrangian or non-conservative form. In addition, Eq. (5) removes the time dependence on $\phi$.

An expression for $\boldsymbol{v_{\mathrm{f}}}$ can be derived by coupling Eqs. (1) and (2) to the momentum conservation equations (e.g., McKenzie, 1984; Bercovici et al., 2001). These are usually solved before Eq. (5) ; a description of the system used in this study is provided below in section 4.1.

### 2.1.1 Numerical Methods

Solving an advection equation using a linear Eulerian scheme leads to high numerical diffusion for first-order schemes, such as the upwind scheme (Courant et al., 1952), and to oscillations on sharp gradients for higher-order schemes (LeVeque, 2002). The latter effect is described by Godunov's theorem (Godunov and Bohachevsky, 1959). This theorem states that linear Eulerian schemes of order of accuracy greater than one cannot preserve the monotonicity of the solution for sharp gradients,
discontinuities or shocks. This has led to extensive developments in the design of high-order Eulerian non-linear schemes that can achieve high accuracy without bringing oscillations. Examples of such developments are the essentially non-oscillatory (ENO) methods (Harten et al., 1987), that later led to weighted essentially non-oscillatory (WENO) schemes (Liu et al., 1994).





These schemes are based on the idea of using a non-linear adaptive procedure to automatically choose the locally smoothest stencil and earlier examples of applications include the modelling of shocks appearing in acoustics (e.g., Grasso and Pirozzoli, 2000) or solving the Hamilton-Jacobi equations (e.g., Jiang and Peng, 2000).

Another approach is to use schemes closer to Lagrangian perspective, such as the Marker-In-Cell (MIC) (or alternatively named Marker-And-Cell) method (e.g., Harlow et al., 1955; Gerya and Yuen, 2003a). It consists at tracking individual particles on a Lagrangian frame and to reinterpolate them when needed on an Eulerian stationary mesh grid. This approach has the advantage of producing little numerical diffusion and to be unconditionnally stable and has been used extensively to advect most fields in geodynamics, such as physical properties and compositions (e.g., Gerya, 2019; van Keken et al., 1997; Duretz et al., 2011).

Finally, there are intermediate methods, such as semi-Lagrangian methods, trying to take advantages from both Eulerian and Lagrangian schemes (Robert, 1981; McDonald, 1984). These schemes look at different particles at each timestep, considering only particles whose final trajectories correspond to the position of a fixed Eulerian grid. This has the advantage of only considering a number of particles equal to the resolution of the Eulerian grid and is computationally efficient. They are also unconditionally stable, but have issues with mass conservation (Chandrasekar, 2022). They were first developed for atmospheric modelling (e.g., Robert, 1981; Staniforth and Côté, 1991) and later successfully used in the plasma modelling community (e.g., Sonnendrücker et al., 1999).

To solve for Eq. (5) in the context of two-phase flow, we implement and test four different advection schemes that are representative of the approaches described above: an upwind scheme, a WENO scheme, a SL scheme and a MIC method.

## 2.2 Upwind scheme

The upwind scheme is among the simplest algorithm for solving an advection equation on an Eulerian grid (e.g., LeVeque, 1992). It is explicit and first-order in space and in time. It consists of using a spatially biased stencil that depends on the direction of the flow (Fig. 1).

### 2.2.1 Spatial discretization

Using a first-order biased spacial stencil, Eq. (5) can be approximated for 1 element and in 1D as:

$$
\begin{aligned}
\frac{\partial C_i}{\partial t} + v_{\mathrm{f},i}\frac{C_i^n - C_{i-1}^n}{\Delta x} = 0 \quad &\text{for} \quad v_{\mathrm{f},i} > 0, \\
\frac{\partial C_i}{\partial t} + v_{\mathrm{f},i}\frac{C_{i+1}^n - C_i^n}{\Delta x} = 0 \quad &\text{for} \quad v_{\mathrm{f},i} < 0,
\end{aligned}
\tag{6}
$$

where $i$ is a spatial index in the $x$ direction and $\Delta x$ is a constant increment in space.

### 2.2.2 Temporal discretization

Combined with the first-order forward Euler method, we retrieve the classical upwind scheme from Eq. (6):





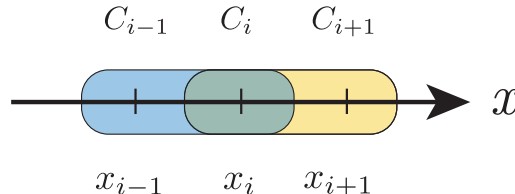

**Figure 1.** Spatial stencil of the Upwind scheme in 1D. The blue box is the valid stencil for positive velocities and the yellow box for negative velocities.

$$\frac{C_i^{n+1} - C_i^n}{\Delta t} + v_{\mathrm{f},i}\frac{C_i^n - C_{i-1}^n}{\Delta x} = 0 \quad \text{for} \quad v_{\mathrm{f},i} > 0,$$
$$\frac{C_i^{n+1} - C_i^n}{\Delta t} + v_{\mathrm{f},i}\frac{C_{i+1}^n - C_i^n}{\Delta x} = 0 \quad \text{for} \quad v_{\mathrm{f},i} < 0,$$

(7)

where $\Delta t$ is the timestep.

It can also be rewritten in a more compact form:

$$C_i^{n+1} = C_i^n - \Delta t\left[v_{\mathrm{f},i}^+\left(\frac{C_i^n - C_{i-1}^n}{\Delta x}\right) + v_{\mathrm{f},i}^-\left(\frac{C_{i+1}^n - C_i^n}{\Delta x}\right)\right],$$

(8)

where

$$v_{\mathrm{f},i}^+ = \max(v_{\mathrm{f},i}, 0),$$
$$v_{\mathrm{f},i}^- = \min(v_{\mathrm{f},i}, 0).$$

This scheme is well known to produce a lot of numerical diffusion and is bounded by the following Courant–Friedrichs–Lewy (CFL) condition, for $p$ dimensions:

$$\Delta t\left(\sum_{j=1}^{p}\frac{|v_{\mathrm{f}_j}|}{\Delta x_j}\right) \leq C_{\max} = 1,$$

(9)

where $C_{\max}$ is the maximum Courant number (e.g., Hirsch, 2007).

### 2.3   Weighted essentially non-oscillatory scheme

Weighted Essentially Non-Oscillatory schemes (WENO) were developed by Liu et al. (1994). The reader can refer to Shu (2009) for a comprehensive review of the development of WENO schemes and Pawar and San (2019) for implementations using Julia.





They are high order schemes able to resolve sharp gradient, produce little numerical diffusion but also follow the same CFL condition as the Upwind scheme. The key idea behind them, is to use a non-linear adaptive procedure to automatically choose

the locally smoothest stencil. This allow WENO schemes to dispose of oscillations when advecting sharp gradients.

We use a 5th order in space finite-difference approach for non-conservative problem, referenced as WENO-5 hereafter.

### 2.3.1  Spatial Discretization

Equation (5) can be discretized in space using WENO-5 scheme similarly to the upwind scheme, in 1D and for 1 element such as:

$$135 \quad \frac{\partial C_i}{\partial t} + v_{f,i}^+ \left( \frac{C_{i+\frac{1}{2}}^L - C_{i-\frac{1}{2}}^L}{\Delta x} \right) + v_{f,i}^- \left( \frac{C_{i+\frac{1}{2}}^R - C_{i-\frac{1}{2}}^R}{\Delta x} \right) = 0, \tag{10}$$

where

$$C_{i+\frac{1}{2}}^L = w_0^L \left( \frac{1}{3}C_{i-2} - \frac{7}{6}C_{i-1} + \frac{11}{6}C_i \right) +$$
$$w_1^L \left( -\frac{1}{6}C_{i-1} + \frac{5}{6}C_i + \frac{1}{3}C_{i+1} \right) +$$
$$w_2^L \left( \frac{1}{3}C_i + \frac{5}{6}C_{i+1} - \frac{1}{6}C_{i+2} \right),$$
$$140 \quad C_{i-\frac{1}{2}}^R = w_0^R \left( -\frac{1}{6}C_{i-2} + \frac{5}{6}C_{i-1} + \frac{1}{3}C_i \right) +$$
$$w_1^R \left( \frac{1}{3}C_{i-1} + \frac{5}{6}C_i - \frac{1}{6}C_{i+1} \right) +$$
$$w_2^R \left( \frac{11}{6}C_i - \frac{7}{6}C_{i+1} + \frac{1}{3}C_{i+2} \right).$$

Here, $C_{i-\frac{1}{2}}^L$ and $C_{i+\frac{1}{2}}^R$ are omitted to avoid redundancy. They can be obtained by shifting the index by -1 and 1 respectively. Non-linear weights $w$ are defined as:

$$145 \quad w_k^L = \frac{\alpha_k}{\alpha_0 + \alpha_1 + \alpha_2}, \quad \alpha_k = \frac{d_k^L}{(\beta_k + \epsilon)^2}, \quad k = 0,1,2$$
$$w_k^R = \frac{\alpha_k}{\alpha_0 + \alpha_1 + \alpha_2}, \quad \alpha_k = \frac{d_k^R}{(\beta_k + \epsilon)^2}. \quad k = 0,1,2$$

The values of the optimal weights $d_k^L$ and $d_k^R$ are given in Table 1 and $\epsilon$ is fixed at $1 \times 10^{-6}$ to avoid division by zero.

Smoothness indicators $\beta$ are equal to:





**Table 1.** Optimal weights for WENO-5 scheme

| $d_k$ | k=0 | k=1 | k=2 |
|-------|-----|-----|-----|
| $d_k^L$ | 0.1 | 0.6 | 0.3 |
| $d_k^R$ | 0.3 | 0.6 | 0.1 |

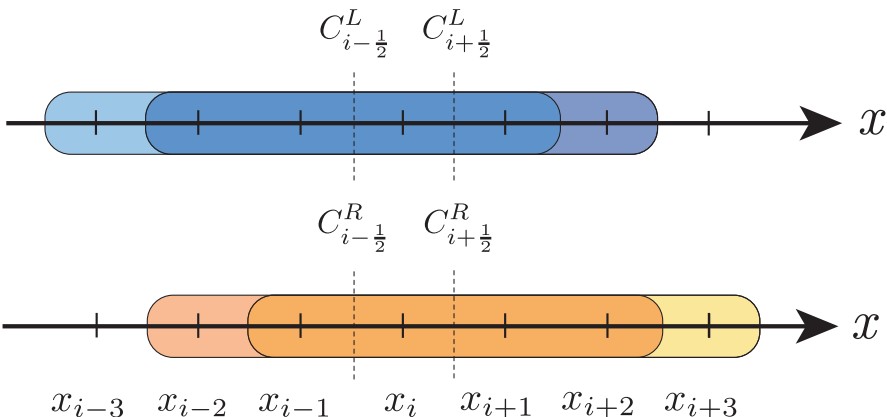

**Figure 2.** Spatial stencil of the WENO-5 scheme in 1D. $C^L$ is used for positive velocities and $C^R$ for negative velocities. The blue boxes are valid stencils for positive velocities and the yellow and orange boxes are valid for negative velocities.

$$\beta_0 = \frac{13}{12}\left(C_{i-2} - 2C_{i-1} + C_i\right)^2 + \frac{1}{4}\left(C_{i-2} - 4C_{i-1} + 3C_i\right)^2,$$
$$\beta_1 = \frac{13}{12}\left(C_{i-1} - 2C_i + C_{i+1}\right)^2 + \frac{1}{4}\left(C_{i-1} - C_{i+1}\right)^2,$$
$$\beta_2 = \frac{13}{12}\left(C_i - 2C_{i+1} + C_{i+2}\right)^2 + \frac{1}{4}\left(3C_i - 4C_{i+1} + 3C_{i+2}\right)^2.$$

WENO-5 in 1D requires a stencil of 5 points biased towards the left for positive velocities and 5 points biased towards the right for negative velocities as shown on Fig. 2. Careful consideration must then be given to boundary conditions for complex problems.

### 2.3.2 Temporal discretization

WENO schemes are not stable using the standard forward Euler time integration method (Wang and Spiteri, 2007). The most commonly used discretisation is the third-order Strong Stability Preserving (SSP) explicit Runge-Kutta method (e.g., Jiang and Shu, 1996; Ghosh and Baeder, 2012). SSP schemes are used to fully capture discontinuous solutions and are therefore very useful for solving hyperbolic partial differential equations (Gottlieb et al., 2001).





The third-order SSP Runge-Kutta for Eq. (5) for 1 element can be written as:

$$C_i^1 = C_i^n - \Delta t L(C_i^n), \tag{11}$$

$$C_i^2 = \frac{3}{4}C_i^n + \frac{1}{4}\left[C_i^1 - \Delta t L\left(C_i^1\right)\right], \tag{12}$$

$$C_i^{n+1} = \frac{1}{3}C_i^n + \frac{2}{3}\left[C_i^2 - \Delta t L\left(C_i^2\right)\right], \tag{13}$$

with $L$ being the spacial discretization operator:

$$L(C_i) = v_{\mathrm{f},i}^+ \left(\frac{C_{i+\frac{1}{2}}^L - C_{i-\frac{1}{2}}^L}{\Delta x}\right) + v_{\mathrm{f},i}^- \left(\frac{C_{i+\frac{1}{2}}^R - C_{i-\frac{1}{2}}^R}{\Delta x}\right).$$

With this formulation, the WENO-5 scheme is fifth order in space and third order in time.

## 2.4    Semi-Lagrangian schemes

Semi-Lagrangian (SL) schemes take a different approach than classical Eulerian methods and are related to tracer-based advection schemes. SL schemes aim to use the best of Lagrangian and Eulerian methods by solving the problem for particles
whose trajectories pass through a fixed grid at the end of each timestep, rather than recording the full history of individual particles. They are therefore unconditionally stable. Two steps are usually required to implement SL schemes: trajectory tracing and interpolation back to the grid. In this study, a backward in time SL scheme is used for the trajectory tracing, and the quasi-monotone scheme developed by Bermejo and Staniforth (1992) for the interpolation.

### 2.4.1    Trajectory tracing

The advantage of backward in time SL schemes is that the interpolant is defined from the Eulerian grid. In the case of a rectilinear grid, this reduces the complexity of the implementation and the numerical cost of the interpolation function. From a particle point of view, the goal is to find the starting points at the previous timestep for each grid point. Using Eq. (5) for 1 element and in 1D, the following equation has to be solved:

$$\frac{dx}{dt} = v_{\mathrm{f}}(x,t). \tag{14}$$

Knowing $x(t_{n+1}) = x_i$ with $i$ a grid point, $x(t_n) = x_d$ with $d$ a departure point needs to be found.

In practice, the velocity field varies greatly in time and in space between each time-step, especially for porosity waves, so it is not easy to determine $x_d$. A common approach to overcome this limit is to use the mid-point scheme (Robert (1981)):

$$\frac{x_i - x_d}{\Delta t} = v_{\mathrm{f}}\left(\frac{x_i + x_d}{2}, t_{n+\frac{1}{2}}\right). \tag{15}$$



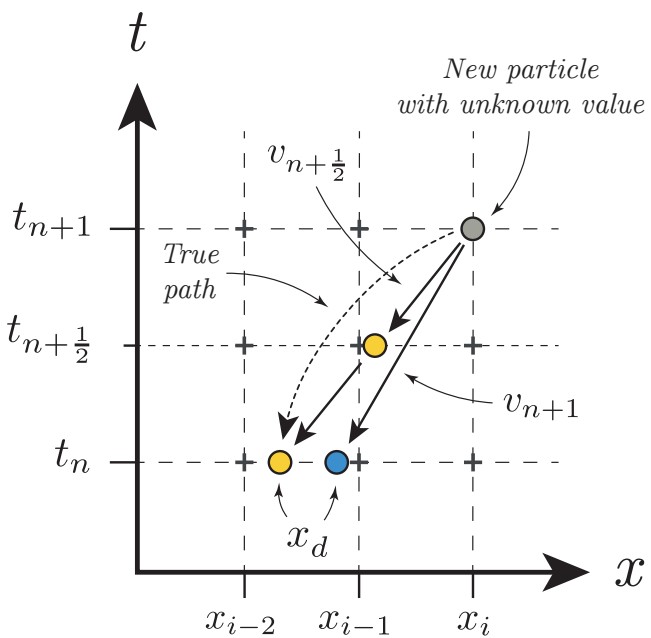

**Figure 3.** Summary of trajectory tracing for backward semi-Lagrangian schemes. The aim is to find the value of the advected quantity at the position $x_i$ and at the timestep $t_{n+1}$. The blue particle uses the velocity at $n+1$. The yellow particle shows the midpoint method, using an approximation of the velocity at $n+\frac{1}{2}$. The value of the particle at position $x_d$ can then be interpolated at $t_n$ to obtain the value at $x_i$ at $t_{n+1}$.

The assumption behind the mid-point rule is that the velocity remains constant at the mid-point value during each timestep.
This ensures that each trajectory is linear, with the mid-point being the average of the positions of its endpoints (Fig. 3). This method is a second-order accurate trajectory method in both space and time.

Equation (15) must be solved implicitly, because $x_d$ is present on both sides of the equation, and therefore requires iterations. It can be achieved for $r$ iterations in that form:

$$x_d^{r+1} = x_i - \Delta t v_{\mathrm{f}} \left( \frac{x_i + x_d^r}{2}, t_{n+\frac{1}{2}} \right).$$ (16)

This also requires interpolation of $v_{\mathrm{f}}$ at $t_{n+\frac{1}{2}}$ $r$ times. A minimum of 3 iterations while using linear interpolation has been shown to be sufficient in most cases (e.g., McDonald, 1984).





### 2.4.2 Interpolation

In most cases, $x_d$ does not correspond to a stencil point on the grid (see Fig. 3). In this case, interpolation is required to retrieve the value of the unknown at $x_d$:

$$C(x_d,t) = \mathcal{L}\left[C(x_{i_k}),t\right], \tag{17}$$

where $\mathcal{L}$ is an interpolation operator and $x_{i_k}$ represents the nodes of the cell containing $x_d$.

Commonly, cubic interpolants are applied as it is a good compromise between performance and accuracy (e.g., Chandrasekar, 2022). The Godunov's theorem still applies to linear SL schemes and since cubic interpolation is third-order in space, it introduces oscillations and overshoots for high gradients. To overcome this limitation, quasi-monotone (QM) SL schemes were developed by Bermejo and Staniforth (1992). The term QM means that the scalar field values cannot exceed the range of the previous timestep, but can still develop wiggles inside that range. Quasi-monotonicity is equivalent to the notion of essentially non-oscillatory (Bermejo, 2001). A disadvantage of this method is an increased numerical diffusion, especially for high Courant number. A maximum Courant number of 1.5 is generally used (e.g., Smith, 2000).

To implement QMSL schemes, let's define $C^-$ and $C^+$ as the minimum and maximum scalar values of the nodes of the cell containing $x_d$ and $C^H$ as the high order non-monotone interpolant. Then, a local clipping can be applied at the end of each timestep:

$$C^M(x_d,t) = \begin{cases} C^+(x_d,t) & \text{if} \quad C^H(x_d,t) > C^+(x_d,t) \\ C^-(x_d,t) & \text{if} \quad C^H(x_d,t) < C^-(x_d,t) \\ C^H(x_d,t) & \text{otherwise}, \end{cases} \tag{18}$$

where $C^M$ is the quasi-monotone interpolant. Equation (18) can be rewritten in a more compact way:

$$C^M(x_d,t) = \min\left\{\max\left[C^-(x_d,t), C^H(x_d,t)\right], C^+(x_d,t)\right\}. \tag{19}$$

Formally, this formulation is equivalent to a linear combination between a high order interpolant and a first order (monotone) interpolant (Bermejo, 2001).

### 2.5 Marker-in-cell schemes

Marker-In-Cell (MIC) schemes share the same ambition as SL schemes, such as being unconditionally stable, but are closer to Lagrangian schemes. They record the complete history of individual particles, called markers and interpolate their values on a fixed grid. This approach has the advantage of reducing greatly numerical diffusion and making MIC schemes unconditionally stable. In addition to trajectory tracing and interpolation, the MIC schemes require markers to be generated within the domain of the model.





### 2.5.1 Initial marker generation and reseeding of particles

The number of markers per cell required can vary depending on the complexity of the problem. Here, 5 markers per cell

dimension, effectively 25 in 2D. This is generally sufficient to achieve good accuracy (e.g., Gerya, 2019). The initial value of each marker can then be obtained by linear interpolation from the initial conditions of the Eulerian grid.

For highly divergent flows, it is necessary to regenerate or remove markers during the simulation, as particles will accumulate in zones with negative divergence values and create a gap in zones with positive divergence values. For reseeding, a strategy similar to Keller et al. (2013) is used. If the marker density per cell is less than 25% of the initial density, new markers

are generated and are assigned the value of the nearest marker. The old markers are discarded after this step. For marker accumulation, the marker density cannot exceed twice the initial density. If it does, a quarter of the markers are discarded at random.

### 2.5.2 Trajectory tracing

The goal of trajectory tracing for MIC schemes is to determine the position of each marker at the next timestep. The same

equation as Eq. 14 is solved. However, compared to backward SL where the final position is known, the unknown in this case is the position of the arrival point. We can rewrite Eqs. 15 and 16 for the unknown $x_a$, the arrival point, using the mid-point rule as:

$$\frac{x_a - x_d}{\Delta t} = v_{\mathrm{f}} \left( \frac{x_a + x_d}{2}, t_{n+\frac{1}{2}} \right), \tag{20}$$

and

235 $$x_a^{r+1} = x_d + \Delta t v_{\mathrm{f}} \left( \frac{x_a + x_d^r}{2}, t_{n+\frac{1}{2}} \right). \tag{21}$$

As for the SL schemes, it is necessary to interpolate the velocity field $r$ times.

Since classical interpolants do not retain the physical properties of the velocity field, such as its divergence, a simple bilinear interpolation may lead to unphysical clustering of markers on the time scale of a numerical model. To address this issue, Pusok et al. (2017) explored different interpolants and showed the advantages of using the so-called LinP interpolation scheme

(Gerya, 2019). The LinP interpolation scheme is an empirical relationship that combines two linear interpolants defined at the sides and at the center of each cell. It is defined as:

$$v_{\mathrm{f}} \left( \frac{x_a + x_d}{2}, t_{n+\frac{1}{2}} \right) = A\mathcal{L} \left[ v_{\mathrm{f}} \left( x_{\mathrm{side}}, t_{n+\frac{1}{2}} \right) \right] + (1 - A)\mathcal{L} \left[ v_{\mathrm{f}} \left( x_{\mathrm{center}}, t_{n+\frac{1}{2}} \right) \right], \tag{22}$$

with $A$, a constant commonly equal to 2/3, $\mathcal{L}$ a linear interpolant, and $x_{\mathrm{side}}$ and $x_{\mathrm{center}}$ the position of the sides and center of the cell containing $x_d$.





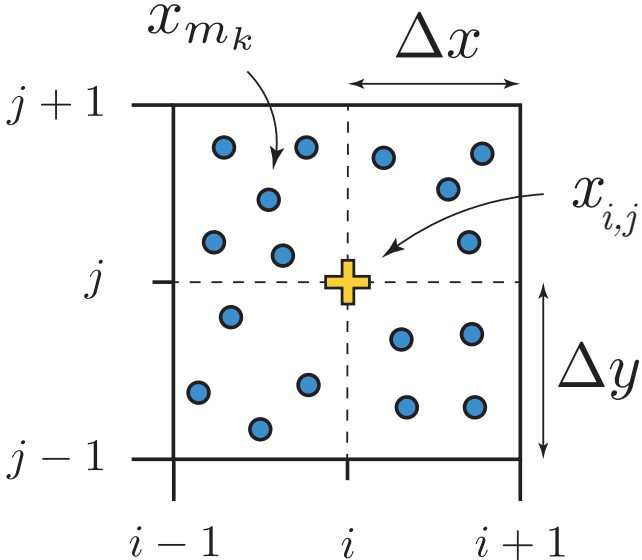

**Figure 4.** Sketch showing the geometric relationship in 2D between a point $x_{i,j}$ of the Eulerian grid and the markers $x_{m_k}$ used for the interpolation on a regular grid. The value at the point $x_{i,j}$ is interpolated from the markers $x_{m_k}$ contained inside the four neighbouring cells. $x_{i,j}$ is fixed in time and in space, whereas the position of the markers $x_{m_k}$ are time-dependent.

### 2.5.3 Interpolation

After calculating the position of the markers, it is necessary to interpolate back on the Eulerian grid. This is more complex than for SL schemes because the markers are not uniformly distributed for a non-trivial velocity field. Therefore, even for regular Eulerian grid, interpolation is performed on an unstructured grid as it is based on the position of the markers.

In this case, interpolation is required to obtain the values of our unknown at $x_i$, where $i$ is a grid point:

$$C(x_i, t_{n+1}) = \mathcal{L}\left[C\left(x_{i_{m_k}}\right), t_{n+1}\right],$$   (23)

where $x_{i_{m_k}}$ represents the surrounding markers used for the interpolant and $\mathcal{L}$ is an interpolation operator. Typically, all markers found in the surrounding cells are used for the interpolation (e.g., Gerya and Yuen, 2003b). Linear interpolants are used because they prevent oscillations, and marker densities are high enough to prevent numerical diffusion. The relationship between the markers and the grid is summarised on Fig. 4 in 2D.

### 3 Numerical test

To test the four advection schemes, the rotation of a cylinder in 2D is performed. The domain is a square of size $1.0 \times 1.0$ with a constant spacing of $\Delta x = \Delta y = 0.005$ for a resolution of $201 \times 201$ nodes. The radius of the cylinder is $24\Delta x$, contains a



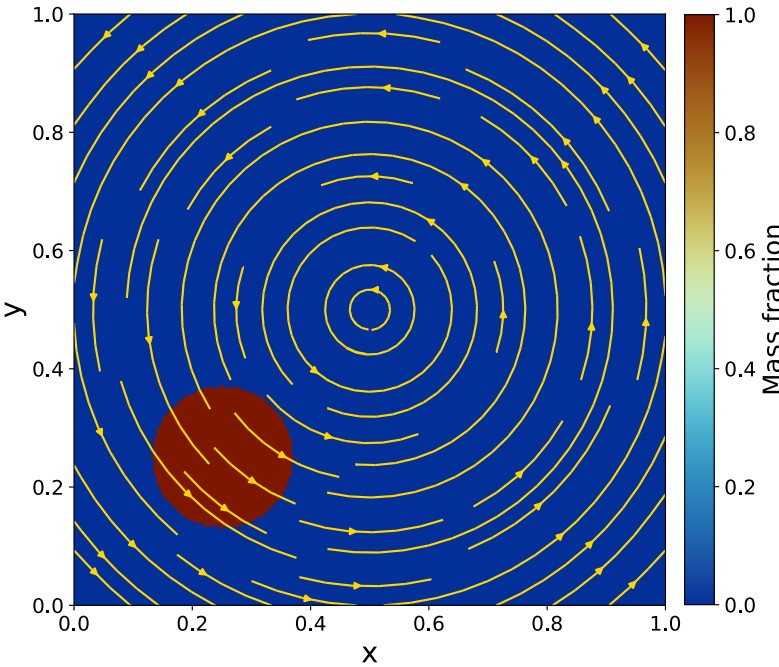

**Figure 5.** Initial conditions for the numerical test consisting of the rotation of a cylinder. The yellow arrows show the velocity field.

mass fraction of 1.0 and is centered at coordinates $(0.25; 0.25)$. The initial conditions are shown on Fig. 5. The time increment $\Delta t$ is set to $400\ s$ with $\omega = \pi \times 10^{-5} s^{-1}$ so that it takes 500 timesteps to make a full revolution. The velocity is defined as

$\boldsymbol{v} = (-\omega(y-0.5); \omega(x-0.5))$ so that the rotation is anti-clock wise and the centre of it is at coordinates $(0.5; 0.5)$. The Courant numbers inside the cylinder range between 0.45 and 0.8. Results after 2 revolutions are shown on Fig. 6 for the four schemes.

To compare and quantify the results of the different schemes, 4 different quantities were monitored: the mass conservation $(M)$, the total error $(E_{\text{tot}})$, the maximum value of the final mass fraction $(\max(C_{\text{f}}))$ and the computational time of 1 timestep $(t_{\text{scheme}})$.

The mass conservation is defined as:

$$M = \frac{\sum_{k=1}^{K}\left(C_{\text{f}}^{k} S_{k}\right)}{\sum_{k=1}^{K}\left(C_{0}^{k} S_{k}\right)}, \tag{24}$$

where $k$ is a 2D grid point index, $K$ is the total number of grid points, $C_{\text{f}}^{k}$ is the final mass fraction at the index $k$, $C_{0}^{k}$ is the initial mass fraction at index $k$ and $S_{k}$ is the surface area of the cell at index $k$.

The total error of the scheme is defined as the mean square error:

$$E_{\text{tot}} = \frac{1}{K}\sum_{k=1}^{K}\left(C_{0}^{k} - C_{\text{f}}^{k}\right)^{2}. \tag{25}$$



**Figure 6.** Results of the rotational test after two revolutions for the upwind, WENO-5, SL QM and MIC schemes (a)-(d). Note that the upwind scheme was run with $\Delta t = 80\,s$ due to stability issues.





**Table 2.** Results of the two numerical tests for four advection schemes.

| Numerical schemes | $M$ | $E_{\text{tot}}$ | $\max(C)$ | $t_{\text{scheme}}$ (ms) |
|---|---|---|---|---|
| Upwind | 1.000 | $31.00 \times 10^{-3}$ | 0.233 | 0.225 |
| WENO-5 | 1.000 | $3.21 \times 10^{-3}$ | 1.001 | 9.856 |
| SL QM | 0.827 | $11.97 \times 10^{-3}$ | 1.000 | 224.450 |
| MIC | 1.000 | $0.68 \times 10^{-3}$ | 1.000 | 710.765 |

The results are reported in Table 2. This shows the strong numerical diffusion of the upwind scheme and its high $E_{tot}$ due to its first order in time and space. WENO-5 shows no oscillation and a good accuracy, being fifth order in space and third in time. SL QM is not mass conservative for this problem but show a relatively good accuracy, being third order in space and second in time and is monotone. Finally, for this test, the MIC scheme is mass conservative, monotone and shows the best accuracy. This simple test highlights the properties of each scheme, but uses a velocity field that is divergence-free and without sharp variations. This is not the case for a real case problem. Coupling with a two-phase flow system is therefore necessary to assert which scheme is the more suitable in this case.

## 4 Coupling chemical advection and two-phase flow

Solving Eq. (5) for concrete cases implies having an expression for $v_{\text{f}}$ at each timestep. In this section, Eq. (5) is coupled to a transport model based on two-phase flow formalism. This transport model is used to model magma ascent in a porous solid phase. The main mechanism of transport is decompaction weakening, buoyancy and failure, and combines the formulations of Connolly and Podladchikov (2007a) and Vasilyev et al. (1998). It considers a compressible visco-elastic matrix with incompressible solid grains and an incompressible fluid phase, and neglects the effect of shear stresses on fluid flow and compaction.

### 4.1 Two-phase Flow Formulation

In the case of a laminar fluid flow, conservation of momentum for the fluid can be expressed using Darcy's law:

$$\phi(\boldsymbol{v_{\mathbf{f}}} - \boldsymbol{v_{\mathbf{s}}}) = -\frac{k(\phi)}{\mu_{\text{f}}}(\nabla P_{\text{f}} + \rho_{\text{f}}\boldsymbol{g}), \tag{26}$$

with $P_{\text{f}}$ the fluid pressure (in Pa), $k$ the permeability (m$^2$), a function of the filled porosity $\phi$, $\mu_{\text{f}}$ the fluid viscosity (Pa.s), and $\boldsymbol{g}$ the gravity vector (m.s$^{-2}$).

The relation between permeability and porosity is assumed to follow the Kozeny-Carman law (Carman, 1939; Costa, 2006):

$$k = a\phi^3, \tag{27}$$

where $a$ is a proportionality constant.





The effective pressure $P_e$ is defined as the difference between lithostatic pressure and fluid pressure:

$$P_e = P_{\text{lith}} - P_{\text{f}}, \tag{28}$$

with $P_{\text{lith}}$ the lithostatic pressure or the vertical load (in Pa). Substituting Eq. (28) in Eq. (26) and assuming constant rock density, we obtain:

$$\phi(\boldsymbol{v_{\text{f}}} - \boldsymbol{v_{\text{s}}}) = \frac{k(\phi)}{\mu_{\text{f}}}(\nabla P_e + \Delta\rho\boldsymbol{g}). \tag{29}$$

Considering the solid phase as a Maxwell body, we introduce rheology as the sum of viscous and poro-elastic deformation:

$$\nabla \cdot \boldsymbol{v_{\text{s}}} = -\frac{P_e}{\zeta(\phi, P_e)} - \phi^b \beta_\phi \frac{\partial P_e}{\partial t}, \tag{30}$$

where $\zeta$ is the volume viscosity of the rock (in Pa.s), $b$, a constant, and $\beta_\phi$ the pore compressibility modulus (Pa$^{-1}$). The terms on the right-hand side represent viscous and poro-elastic deformation, respectively. Equation (30) is valid on the basis that shear stress is neglected.

The volume viscosity $\zeta$ is defined as a function of $\phi$ and $P_e$:

$$\zeta = \frac{\mu_{\text{s}}}{\phi^m}[\frac{1}{R} - H(P_e)(\frac{1}{R} - 1)], \tag{31}$$

with $\mu_{\text{s}}$ the shear viscosity of the rock (in Pas), $m$ a constant, and R the decompaction weakening factor, defined as the inverse of the R factor in Connolly and Podladchikov (2007a). $H(P_e)$ is originally defined as the Heaviside function but is here approximated by a hyperbolic tangent function as similarly done by Räss et al. (2018).

We approximate here $\beta_\phi$ as the inverse of $G$, the shear modulus of the rock (in Pa):

$$\beta_\phi \approx \frac{1}{G}. \tag{32}$$

This is valid for cylindrical pores, as described by Yarushina and Podladchikov (2015).

Summing up the right-hand sides of Eqs. (1) and (2) describing mass conservation and neglecting the change of densities, we obtain the total volumetric flux of material. Applying the divergence operator, we can derive:

$$\nabla \cdot [\boldsymbol{v_{\text{s}}} + \phi(\boldsymbol{v_{\text{f}}} - \boldsymbol{v_{\text{s}}})] = 0. \tag{33}$$

We can then substitute Eqs. (29) and (30) in Eq. (33) to obtain:

$$\frac{P_e}{\zeta(\phi, P_e)} + \frac{\phi^b}{G}\frac{\partial P_e}{\partial t} = \nabla \cdot \left[\frac{k(\phi)}{\mu_{\text{f}}}(\nabla P_e + \Delta\rho\boldsymbol{g})\right]. \tag{34}$$





In addition, developing Eq. (1) with the assumption that $\phi$ is much smaller than unity, and substituting with Eq. (30) yields:

$$\frac{\partial \phi}{\partial t} = -\frac{P_e}{\zeta(\phi, P_e)} - \frac{\phi^b}{G}\frac{\partial P_e}{\partial t}.$$

(35)

Equation (34) can be seen as the mass conservation equation of the system, relating the flux densities of the solid and fluid phases. Equation (35) relates the evolution of porosity with the deformation of the solid phase.

Solving these two coupled equations for $P_e$ and $\phi$ allows the calculation of $\boldsymbol{v_s}$ and $\boldsymbol{v_f}$ from Eqs. (30) and (29) at each timestep, making the link with Eq. (5).

## 4.2 Nondimensionalization and numerical approach

To mitigate numerical errors, a dimensionless scaling of the system is applied. We define the following scaling variables (McKenzie, 1984; Connolly and Podladchikov, 1998):

the characteristic porosity $\phi^*$:

$$\phi^* = \phi_0,$$

(36)

with $\phi_0$, the background porosity.

The characteristic viscous compaction length $L^*$:

$$L^* = \sqrt{\frac{\mu_s k_0}{\phi_0 \mu_{f0}}},$$

(37)

with $k_0$, the background permeability (in m$^2$).

The characteristic effective pressure $P_e^*$:

$$P_e^* = L^* \Delta \rho g.$$

(38)

The characteristic fluid flux $q_f^*$:

$$q_f^* = \frac{\mu_{f0}}{\phi_0} \Delta \rho g L^*.$$

(39)

The characteristic time $t^*$:

$$t^* = \frac{L^*}{\sqrt{q_f^*}}.$$

(40)





$De$, the Deborah number:

$$De = \beta_\phi P_e^*. \tag{41}$$

$De$ is formally the ratio of the relaxation time to the observation time (Reiner, 1964), and here characterises the ratio between viscous and elastic deformation.

Using these variables with Eqs. (29), (30), (34) and (35), and rearranging, we obtain the dimensionless system of equations:

$$\frac{\partial p}{\partial t} = \frac{1}{\varphi^b De} \times \left( \nabla \cdot [\varphi^n (\nabla p + 1)] - \frac{\varphi p}{\frac{1}{R} - H(p)(\frac{1}{R} - 1)} \right), \tag{42}$$

$$\frac{\partial \varphi}{\partial t} = - \left[ \frac{\varphi p}{\frac{1}{R} - H(p)(\frac{1}{R} - 1)} + \varphi^b De \frac{\partial p}{\partial t} \right], \tag{43}$$

$$\nabla \cdot \boldsymbol{u_s} = \frac{\partial \varphi}{\partial t}, \tag{44}$$

$$\boldsymbol{u_f} = \varphi^{n-1} (\nabla p + 1) + \boldsymbol{u_s}, \tag{45}$$

where $\varphi$, $p$, $\boldsymbol{u_s}$ and $\boldsymbol{u_f}$ are the dimensionless porosity, the dimensionless effective pressure, the dimensionless solid velocity and the dimensionless fluid velocity, respectively.

Equations (42) and (43) are strongly coupled and highly stiff due to the nonlinearity of the system and require an efficient numerical solver. DifferentialEquations.jl (Rackauckas and Nie, 2017), a robust ordinary differential equation (ODE) solver, was used. This method has the advantage of simplicity, both in concept and in coding, and allows arbitrary orders of accuracy

in time to be easily tested using different ODE solvers.

Equations (42) and (43) are first discretised in space using finite differences on a uniform Cartesian grid in 2D, and then integrated in time using the Trapezoidal Rule with the second order Backward Difference Formula (TR-BDF2) scheme, an implicit scheme suitable for highly stiff problems (Bank et al., 1985). It uses adaptive time-stepping and Newton method as a non-linear solver, using automatic differentiation to compute the Jacobian matrix. Knowing $\varphi$ and $p$, Eqs. (44) and (45) are

then solved to compute $\boldsymbol{u_s}$ and $\boldsymbol{u_f}$ at each timestep. The boundary conditions are periodic in all directions for all models. The system is then dimensionalised back.




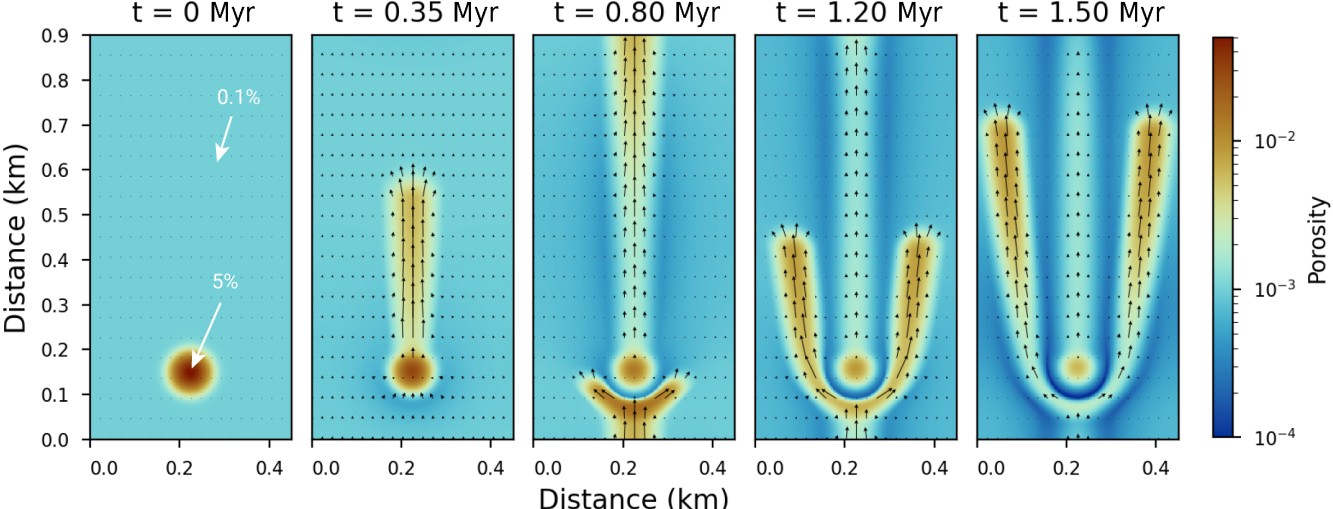

**Figure 7.** Reference evolution of the porosity in a 2D model from an initial Gaussian anomaly, which forms porosity waves. The superimposed vector field shows the melt velocity. Periodic boundaries are applied on all sides. The initial porosity anomaly is a Gaussian function with a maximum value of 5%. The background porosity is 0.1%. The spatial resolution of the grid is 300×600. The physical parameters used are listed in Table 3. The melt velocity is scaled by relative magnitude.

### 4.3 Application to magmatic system

To assert the behaviour of the four advection schemes coupled with a two-phase flow system, we model the ascent of a magmatic anomaly. The spatial domain is a 2D regular grid of 450 by 900 m and the total physical time is 1.5 Ma. The initial melt fraction distribution is defined using the following 2D Gaussian function:

$$\phi = \phi_0 + \phi_{max} \times \exp\left(-\frac{(x-x_0)^2 + (z-z_0)^2}{\sigma^2}\right), \tag{46}$$

with $\phi_0$, the background porosity defined as 0.1%, $\phi_{max}$, the maximum porosity defined as 5%, and $x_0$ and $z_0$, the center of the anomaly. $\sigma$ is the standard deviation of the Gaussian and is 30. All physical parameters and corresponding scaling variables used are reported in Table 3. The evolution of porosity is shown on Fig. 7. All models were performed on a single computer with an Intel Xeon Gold 6128 processor and 128 GB of RAM.

The melt fraction is linked with two different arbitrary chemical compositions: a basaltic composition for the background melt fraction and an andesitic composition for the anomaly. The goal is not to model a realistic magmatic system, but to investigate how the advection schemes can numerically affect the predictions of the model. The two compositions are reported in Table 4. No feedback between the melt compositions and the physical properties of the melt were considered, to prevent the advection schemes to influence on the two-phase flow. In nature, the melt composition has an effect on the melt viscosity and density. The maximum timestep allowed for the two-phase flow is constrained by the Courant number associated with the





**Table 3.** Parameters and corresponding scaling variables used in the models.

| Parameter | Symbol | Value | Unit |
|---|---|---|---|
| Melt viscosity | $\mu_f$ | 100 | Pa.s |
| Rock shear viscosity | $\mu_s$ | $10^{19}$ | Pa.s |
| Density contrast between solid and melt | $\Delta\rho$ | 500 | kg.m$^{-3}$ |
| Weakening parameter | $R$ | 100 | |
| Shear modulus | $G$ | $3.5 \times 10^{19}$ | Pa |
| Background porosity | $\phi_0$ | $10^{-3}$ | |
| Exponent for bulk viscosity term | $m$ | 1 | |
| Exponent for poro-elastic term | $b$ | 1 | |
| Permeability constant | $a$ | $10^{-7}$ | |
| Gravity acceleration | $g$ | 9.81 | m.s$^{-2}$ |
| Characteristic viscous compaction length | $L^*$ | 100 | m |
| Characteristic effective pressure | $P_e^*$ | 490332.5 | Pa |
| Characteristic time | $t^*$ | 0.65 | Myr |
| Characteristic fluid flux | $q_f^*$ | $4.9 \times 10^{-12}$ | m.s$^{-1}$ |
| Deborah Number | $De$ | $1.4 \times 10^{-5}$ | |

**Table 4.** Melt compositions in wt% used in the models. 1: Recalculated from Giordano and Dingwell (2003). 2: Recalculated from Neuville et al. (1993).

| Oxide (wt%) | Basalt[1] | Andesite[2] |
|---|---|---|
| $SiO_2$ | 48.32 | 59.87 |
| $TiO_2$ | 1.65 | 0.82 |
| $Al_2O_3$ | 16.72 | 16.93 |
| FeO(T) | 10.41 | 5.28 |
| MgO | 5.31 | 3.28 |
| CaO | 10.75 | 5.70 |
| $Na_2O$ | 3.85 | 3.76 |
| $K_2O$ | 1.99 | 1.36 |
| $H_2O$ | 1.00 | 3.00 |

melt velocity. Its maximum value allowed for upwind and WENO-5 is 0.7 but 1.5 for QMSL and MIC to take advantage of the extended stability of these schemes. The results for the evolution of the silica content in the melt are shown on Fig. 8.

As there is no analytical solution to this particular problem, it is not possible to directly calculate the numerical error of the different advection schemes. Nevertheless, we can compute the mass conservation of the advected quantities. The total mass of the melt composition is conserved, as it is renormalized to 100% at each timestep. However, it is not necessary the case for



**Figure 8.** Evolution of the silica content in the melt for four different advection schemes: upwind, WENO-5, QMSL and MIC (a)-(d). The Gaussian anomaly of porosity is associated with an andesitic composition, whereas the background porosity has a basaltic composition. The corresponding two-phase flow has an adaptive timestep limited to a maximum value of Courant number below 0.7 for the upwind and WENO-5 schemes and below 1.5 for QMSL and MIC. The spatial resolution is $300 \times 600$ nodes. The physical parameters used for the two-phase flow are reported in Table 3 and are identical for all models.





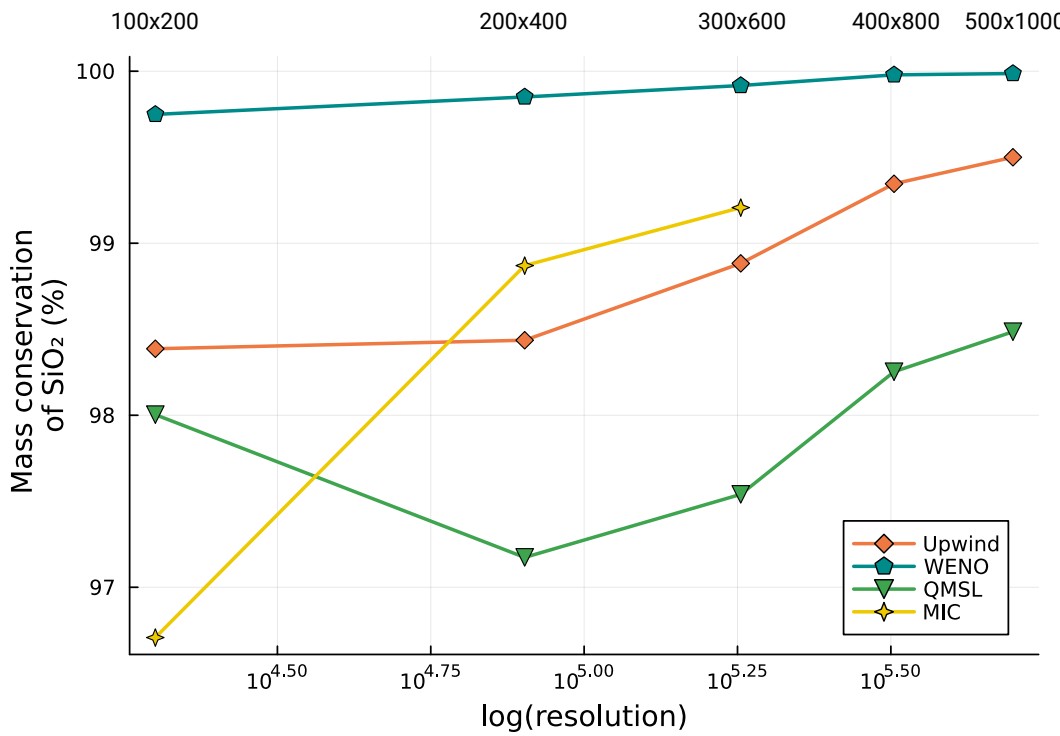

**Figure 9.** Mass conservation of silica content in the melt fraction for four different advection schemes and five different spatial resolutions. The Courant number used is 0.7 for WENO-5 and upwind and 1.5 for QMSL and MIC. The resolutions are $100\times200$, $200\times400$, $300\times600$, $400\times800$ and $500\times1000$. MIC models for $400\times800$ and $500\times1000$ were not performed due to RAM memory saturation. The physical parameters used are reported in Table 3.

each individual oxide. In that light, similar to Eq. (24), we monitor the mass conservation of $SiO_2$ ($M_{SiO_2}$) in the melt at each timestep:

$$M_{SiO_2} = \frac{\sum_{k=1}^{K}\left(\phi^k C_{SiO_2}^k S_k\right)}{\sum_{k=1}^{K}\left(\phi_0^k C_{SiO_{2_0}}^k S_k\right)}, \tag{47}$$

where $\phi^k$ and $\phi_0^k$ are the current and initial porosity at index $k$, and $C_{SiO_2}^k$ and $C_{SiO_{2_0}}^k$ the current and initial $SiO_2$ composition in the melt at the index $k$.

The melt fraction $\phi$ is conserved through the models, as Eqs. (42) and (43) are solved using a conservative discretisation. Therefore, $M_{SiO_2}$ only monitors the effects of the advection schemes. To quantify how the mass conservation evolves, the same model was performed at five different resolutions: $100\times200$, $200\times400$, $300\times600$, $400\times800$ and $500\times1000$. The values

of $M_{SiO_2}$ for each resolution are shown on Fig. 9. The running time of each model is reported on Fig. 10.



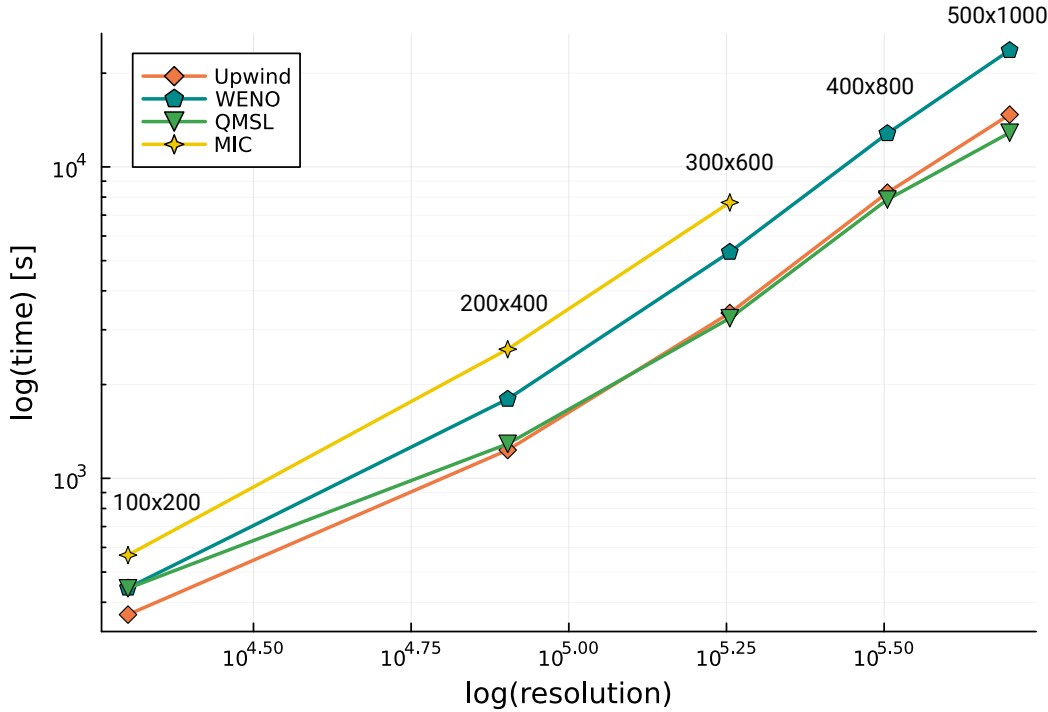

**Figure 10.** Total running time of two-phase flow system coupled with four different advection schemes and five different resolutions. The Courant number used is 0.7 for WENO-5 and upwind and 1.5 for QMSL and MIC. The resolutions are 100×200, 200×400, 300×600, 400×800 and 500×1000. MIC models for 400×800 and 500×1000 were not performed due to RAM memory saturation. The physical parameters used are reported in Table 3.

## 5 Results and discussion

The numerical models produced allow a better understanding of the process of passive chemical transport in magma within porosity waves and the impact of each advection scheme on the magma composition over time. All models confirm two distinct composition domains at the top of the porosity waves at the end of the simulations (Fig. 8). It is effectively a mixture of the compositions from the initial background porosity and from the anomaly. This is due to melt being incorporated by the waves as they rise, and has also been reported in previous studies (e.g., Jordan et al., 2018).

Comparing the results of the four algorithms, it is clear that the upwind scheme has the highest amount of numerical diffusion and the lowest accuracy, which increases chemical mixing for non-physical reasons. WENO-5 and QMSL exhibit similar results in terms of numerical diffusion while MIC shows the lowest amount with almost purely advective behaviour (Fig. 8). This is consistent with the numerical tests (Fig. 6). In terms of mass conservation, the silica content is not conserved in all four schemes (Fig. 9). WENO-5 gives the best results, with a mass conservation ranging from 99.75 to 99.99% from the lowest to the highest resolution. MIC and upwind show values around 98.5 to 99.5% at medium resolution, and QMSL





exhibits significantly lower values, ranging from 97.17 to 98.49%. Concerning performance, two factors affect the results: the stability of the advection schemes, that is linked on how often the two-phase flow solver is called, and the performance of the advection schemes themselves (Fig. 10). QMSL shows the best scaling as it has both an extended stability domain and a low computational cost. upwind shows similar performance despite being bounded by the Courant condition. This is due to its very simple algorithm. WENO-5 has a higher computational cost, up to twice that of QMSL and upwind at high resolution. Finally, MIC is the most computationally expensive, both in terms of memory and running time, although it has an extended stability domain. All the calculations were performed on a single core and the code could be further optimised, but this provides an idea of the cost of each method. For large-scale computations, the extended stability domain may be the dominant criterion for efficient computation, as all the advection schemes are explicit, don't require iterations and are easily parallelizable. The two-phase flow solver will then be the limiting factor in computational efficiency.

The upwind scheme is considered inadequate for this problem because of its high numerical diffusion and low accuracy. The MIC scheme shows very good results in terms of accuracy with the least amount of numerical diffusion and has no stability condition. However, it lacks mass conservation and is expensive in terms of computation and memory, since it needs to keep track on the markers. As the velocity field $v_f$ is strongly divergent, it requires to frequently regenerate and delete markers, which adds complexity to the implementation and additional numerical cost. As a result, we consider this scheme to be too costly for this particular problem. In addition, the QMSL scheme shows very good performance with its extended stability field and good accuracy, but has very poor mass conservation. Attempts have been made to make SL schemes mass conservative, either globally (e.g., Bermejo and Conde, 2002; Zerroukat et al., 2002) or locally (e.g., Qiu and Shu, 2011). These methods require the conservative form of the advection equation, a global correction term or are only applicable to divergent free vector fields. The QMSL scheme could be a good candidate for problems where mass conservation is not critical, such as advection of solid properties or solving the advection part of the heat equation, still in the context of two-phase flow problems but is not suitable for chemical advection where mass conservation is important.

Based on these results, the WENO-5 advection scheme appears to perform best for this problem. Mass conservation is a critical property for studying mass balance and mass transport problems linked to magma transport at different scales on Earth and this algorithm obtains the best results. It also has good accuracy, reasonable performance and is easy to extend to higher dimensions. If complex boundary conditions are required, it is possible to reduce the 5 point spatial stencil to a 3 point stencil and simplify the implementation of the boundary conditions, using the Compact-Reconstruction WENO-5 (CRWENO-5) scheme (Ghosh and Baeder, 2012). This approach uses implicit candidate stencils, which makes parallel computing more difficult but not impossible (Ghosh et al., 2015).

## 6 Conclusions

In this study, a series of test was performed to determine which advection schemes is the most suitable for modelling chemical transport of a magma, we compared four of the most commonly used in the literature: the upwind, WENO-5, MIC, and QMSL



schemes. To test them, we combined a 2D two-phase flow model, which describes the evolution of the melt fraction of a magma over time, with the chemical advection of its composition.

All algorithms, except the upwind scheme, are able to predict the melt composition with reasonable accuracy. However, mass conservation of each individual oxide in the melt is not fully achieved. Nevertheless, WENO-5 has the lowest amount of mass loss, even at low resolution, is explicit, easy to implement and to extend in 3D, although it is bound by the CFL condition.

On the basis of these results, the WENO-5 scheme is the most appropriate to use for transporting magma composition during magma ascent. This is also valid for problems using similar formulations, such as chemical advection in aqueous fluids, and makes WENO-5 a suitable scheme for modelling reactive transport.

*Code availability.* The last version of the code used in this study allowing reproducibility of the data is available on GitHub at (https://github.com/neoscalc/ChemicalAdvectionPorosityWave.jl). Past and future versions are available at a permanent DOI repository (Zen-

440 odo) at: https://doi.org/10.5281/zenodo.8411354. The code is written in the Julia programming language. Refer to the repository's README for additional information. The code is distributed under the GPL-3.0 license.

*Author contributions.* HD and PL conceptualised the project. PL acquired the funding. HD conducted the study with input from NR. HD wrote the code and the original manuscript. NR and PL revised and edited the manuscript. All authors contributed to the discussions and interpretation of the results.

*Competing interests.* The contact author has declared that none of the authors have competing interests.

*Acknowledgements.* Funding was provided by the European Research Council (ERC) under the European Union's Horizon 2020 research and innovation programme (grant agreement No 850530).



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
