# Peer review of "Modelling chemical advection during magma ascent"

_Geoscientific Model Development, 2023_

## Referee Comment (RC1)

I find the manuscript entitled "Modelling chemical advection during magma ascent" by Dominguez et al. quite interesting and helpful for the geoscience community. In my view, the problem of numerical advection is often underappreciated, and the presented study clearly shows the pitfalls of using too simplistic advection schemes. In their work, the authors introduce and compare the performance of four selected numerical advection schemes. In conclusion, they advocate the usage of the WENO type of schemes for petrological applications (chemical advection). Below I give my general and detailed suggestions that the authors may consider to follow before the final publication of the manuscript.

The numerical schemes selected for the study are introduced as simplified 1D variants and later tested using 2D setups. Although the 2D generalizations are rather straight-forward, in my view, it would be worth to give some clues to the readers on how they are developed. In particular, the 2D generalization of the WENO scheme that builds on a 5-point stencil in 1D is perhaps least obvious in this respect (not to mention the treatment of the boundary conditions). In fact, in the final parts of the manuscript, the authors themselves mention an alternative formulation for the 2D WENO scheme that builds on a 3x3 stencil.

I was a bit confused about the MIC scheme that was used in the study. The authors mention the initial step of interpolating the field values from grid nodes to markers. Considerable focus was given to integrating the trajectories of the markers. The step of interpolating the field values back from markers to nodes was sketched out. Unless I got it wrong, the advected fields in this study are essentially phase fields, so no calculations and updates need to be performed on the Eulerian grid. Thus, there is no need to interpolate back these updates to markers in such cases. However, interpolating the base field values from grid nodes to markers would be an undesired and suboptimal step leading to unnecessary numerical diffusion. I think that it would be helpful to clarify the design of the MIC scheme that was used in the tests. What is the reason for the observed mass loss in the second test using the MIC scheme? Could it be related to the non-conservative reseeding and removal of the markers? I would suspect that the mass conservation of the standard MIC should be almost ideal.

The author presented the rigid-body rotation test as their base test. In my view, a bit more complex synthetic 2D test such as for example the shear cell test would be a great addition to the study. On a different note, I actually wonder how to design a balanced comparison of the accuracy of grid-based advection schemes against marker-based methods, when their numerical resolutions largely differ (for a fixed grid resolution, a much larger number of markers is used than nodes in this study). In addition, the authors took advantage of the extended stability of the MIC and SL schemes and used a larger value of the time step than in the case of the upwind and WENO schemes, however, this may have an effect on the numerical performance of the studied schemes.

I was confused by the fact that it was not possible to obtain MIC results using a 128 GB machine for grid resolutions such as 500x1000, when the number of markers is on the order of 10 million. What was the exact reason for such limit in the current MIC implementation? The final conclusions build on the statement that the MIC scheme lacks mass conservation and it is expensive in terms of computation and memory. I already expressed my concerns about the observed lack of mass conservation in the case of the MIC scheme. I think that it would be also

useful if the authors could elaborate more on the computational and memory performance of their MIC implementation.

Detailed comments:

l. 38 What is exactly the notion of triviality in this case?

l. 43 "This brings limitation to the resolution of the model …" - It is indeed correct that for any given computational resources 1D numerical models can be studied with a higher resolution than 2D models. On the other hand, looking from a 3D physical perspective using approximate 2D numerical models could be considered as relaxing computational limitations to the resolution of the model.

eq.5 Is it necessary to use the brackets around $C_e^f$?

l. 84 Perhaps early rather than earlier?

l. 87 "It consists at tracking individual particles on a Lagrangian frame and to reinterpolate them when needed on an Eulerian stationary mesh grid". consist at or consist of? to reinterpolate or reinterpolating?

l. 94 "..to the position of a fixed Eulerian grid.." It would be perhaps worth to mention grid nodes here.

l. 106 Please change spacial to spatial.

l. 106 It might be a bit confusing when the term "element" is mentioned here.

eq. 6 It would be perhaps more transparent if the f symbol, which stands for fluid, was used in the superscript rather than subscript. It could be mentioned that v denotes the x component of the velocity.

l. 109 Please consider using "grid spacing" rather than "increment in space".

eq. 9 Please consider using the f symbol in the superscript. The j index that denotes the spatial component of the velocity could be then used in the subscript.

l. 129 Here and in fig. 1 caption, "Upwind" is capitalized, while it is not in l. 111.

l. 129 Please consider removing the comma.

l. 133 Perhaps "for a single grid node" rather than "1 element" would suit better here. This issue reappears in other parts of the manuscript.

l. 153-54: What is meant by "complex problems" here? I suspect that for any problem boundary conditions need a special or careful treatment when a 5-point stencil is used. I don't

think that it is necessary to present in details how BC are treated in this case, but a short description would be useful.

l. 158 It is not that important, but I'm wondering about the notion of "fully capture" in this case.

l. 164 Please replace spacial by spatial.

l. 171 What is "therefore" referring to?

l. 182 " … is not easy to determine xd. A common approach to overcome this limit…" In my vewi, the term limit may be misleading in this context. I would suggest something like: A common approach to accurately determine …

l. 193 "a stencil point on the grid" Perhaps it could be described as "a grid node".

l. 196 It could be several (neighboring) cells depending on the interpolation order (here cubic, so four nodes are needed to span the interpolation space). On the other hand, it depends on the adopted definition of the cell. Anyway, I think that it would be worth to clarify this issue.

l. 214 "…interpolate their values…". Perhaps "…interpolate field values associated with them…"

l. 220 "The initial value of each marker …" Perhaps "The initial field value in each marker …"

l. 222-23 The problem of deteriorating marker resolution is not only limited to highly divergent flows. It may also affect incompressible, but strongly stretching flows. In such cases, the density of markers can actually depend on the direction (markers may be locally jammed in a certain direction and rarefied in the direction that is perpendicular). Thus, simple scalar measures of marker density may fail in such cases.

L. 246-54 In my view, based on the presented description, it is not exactly clear how the field values are interpolated from markers to grid nodes. Is it the least-square type of linear interpolation? How many surrounding cells are used?

l. 256-261 The domain size is given as dimensionless, while the time is in seconds.

l. 268 Isn't $S_k$ constant for all the internal cells?

l. 352-354 Is this the description of the ODE solver implemented in DifferentialEquations.jl (l. 348)? Please clarify.

l. 363 The standard deviation is given as a dimensionless quantity, while the size of the model domain is given in meters.

l. 366-67 I might be missing something here, but I don't exactly understand how the initial distribution of the melt chemistry is governed. In fig. 8 it appears just bi-modal at t=0 Ma.

Table 3 Please fix the value of the shear modulus (an incorrect exponent is used).

l. 374-383 This part might be a bit confusing. Although the link to eq. 5 is mentioned in l. 320, I think that it would be useful to clarify which physical fields are advected in this model. Please explain why the oxide content may not be conserved, despite the melt fraction is claimed to be conserved due to the chosen numerical scheme. If the total mass of the melt composition (summed weight percentages?) is renormalized in each time step, then I'm not sure if it could be treated as a conserved quantity.

l. 389 What is meant by mixture here?

l. 397 I am a bit confused by the observed lack of mass conservation in the upwind scheme. Is it due to the fact that the renormalization procedure is used? I would also expect MIC to be essentially mass conserving and non-diffusive. Are the observed mass loss effects due to marker reseeding and removal?

l. 407 Using mid-point for computing trajectories in MIC and SL results in some iterations.

l. 423 What is exactly meant by complex boundary conditions?

Fig. 9 & 10 The axes are logarithmic and there is no need to add log(.) in the labels.

---

## Referee Comment (RC2)

[revised manuscript text omitted]

Solving these two coupled equations for $P_e$ and $\phi$ allows the calculation of $v_{\mathbf{s}}$ and $v_{\mathbf{f}}$ from Eqs. (30) and (29) at each
320   timestep, making the link with Eq. (5).

**4.2   Nondimensionalization and numerical approach**

To mitigate numerical errors, a dimensionless scaling of the system is applied. We define the following scaling variables
(McKenzie, 1984; Connolly and Podladchikov, 1998):
the characteristic porosity $\phi^*$:

325   $$\phi^* = \phi_0, \tag{36}$$

with $\phi_0$, the background porosity.

The characteristic viscous compaction length $L^*$:

$$L^* = \sqrt{\frac{\mu_{\mathrm{s}} k_0}{\phi_0 \mu_{\mathrm{f0}}}}, \tag{37}$$

with $k_0$, the background permeability (in m$^2$).
330   The characteristic effective pressure $P_e^*$:

$$P_e^* = L^* \Delta \rho g. \tag{38}$$

The characteristic fluid flux $q_{\mathrm{f}}^*$:

$$q_{\mathrm{f}}^* = \frac{\mu_{\mathrm{f0}}}{\phi_0} \Delta \rho g L^*. \tag{39}$$

[revised manuscript text omitted]

---

## Author Comment (AC1)

I find the manuscript entitled "Modelling chemical advection during magma ascent" by Dominguez et al. quite interesting and helpful for the geoscience community. In my view, the problem of numerical advection is often underappreciated, and the presented study clearly shows the pitfalls of using too simplistic advection schemes. In their work, the authors introduce and compare the performance of four selected numerical advection schemes. In conclusion, they advocate the usage of the WENO type of schemes for petrological applications (chemical advection). Below I give my general and detailed suggestions that the authors may consider to follow before the final publication of the manuscript.

The numerical schemes selected for the study are introduced as simplified 1D variants and later tested using 2D setups. Although the 2D generalizations are rather straight-forward, in my view, it would be worth to give some clues to the readers on how they are developed. In particular, the 2D generalization of the WENO scheme that builds on a 5-point stencil in 1D isperhaps least obvious in this respect (not to mention the treatment of the boundary conditions). In fact, in the final parts of the manuscript, the authors themselves mention an alternative formulation for the 2D WENO scheme that builds on a 3x3 stencil.

Response: The extension of the WENO scheme to 2D is rather similar to the extension of an upwind scheme to 2D, where each flux term for each dimension is simply calculated independently and summed up at the end before the time stepping procedure, so that each dimension can be treated independently as described in 1D.

→ Explanations have been added in the revised manuscript.

The mention of the compact WENO was mention to give an alternative to readers who are not interested in using parallelizable code for less computational expensive applications, as it reduces to the use of one ghost point for the boundaries. Also, the mention of the compact WENO reconstruction is not something specific to 2D or 3D but is also applicable to 1D. It is simply an alternative discretization that use an implicit stencil, hence allowing smaller stencils. The cost is that this requires to solve a linear system at each timestep, which is not easily parallelizable. Both reviewers shown that this was confusing, so the mention of the compact WENO scheme has been removed from the manuscript.

→ The notion of compact WENO has been removed from the conclusion.

I was a bit confused about the MIC scheme that was used in the study. The authors mention the initial step of interpolating the field values from grid nodes to markers. Considerable focus was given to integrating the trajectories of the markers. The step of interpolating the field values back from markers to nodes was sketched out. Unless I got it wrong, the advected fieldsin this study are essentially phase fields, so no calculations and updates need to be performedon the Eulerian grid. Thus, there is no need to interpolate back these updates to markers in such cases. However, interpolating the base field values from grid nodes to markers would bean undesired and suboptimal step leading to unnecessary numerical diffusion. I think that it would be helpful to clarify the design of the MIC scheme that was used in the tests. What is the reason for the observed mass loss in the second test using the MIC scheme? Could it be related to the non-conservative reseeding and removal of the markers? I would suspect that the mass conservation of the standard MIC should be almost ideal.

Response: in the tests, there is indeed no need to reinterpolate between the markers and the grid as a passive advection is performed. Nevertheless, the focus of this paper is to find the best

solution for advecting the chemical composition of a melt, with the ultimate goal of moving towards reactive transport modelling, as it is mentioned in the introduction of this paper. With this in mind, the authors believe that it is relevant to include a reinterpolation step, since in fully coupled system, the change in viscosity and density of the melt induced by the change in chemical composition will have an important effect on the two-phase flow. As this step is not required in the other numerical schemes as they are performed at the grid points, the authors believe that it is relevant to add this step for a fairer comparison between the different advection schemes.

→ This was mentioned in the manuscript.

Concerning the mass loss observed in the MIC scheme for the coupling with the two-phase flow, there was a mistake in the implementation of the scheme (as well as for the QMSL). The results after correction show way less mass loss, probably reflecting more what the reviewer has in mind. This mass loss can be indeed attributed to non-conservative reseeding and removal of the markers.

→ New results after correction of the MIC implemented were added to the manuscript

The author presented the rigid-body rotation test as their base test. In my view, a bit more complex synthetic 2D test such as for example the shear cell test would be a great addition to the study. On a different note, I actually wonder how to design a balanced comparison of the accuracy of grid-based advection schemes against marker-based methods, when their numerical resolutions largely differ (for a fixed grid resolution, a much larger number of markers is used than nodes in this study). In addition, the authors took advantage of the extended stability of the MIC and SL schemes and used a larger value of the time step than in the case of the upwind and WENO schemes, however, this may have an effect on the numerical performance of the studied schemes.

Response:

→ A new 2D test was added in the revised manuscript, a convection test using a parametrised equation for the velocity field simulating the advection in a single convection cell.

The question on how to properly compare fundamentally different schemes is indeed important. It is fairly straightforward to determine the accuracy of each scheme depending on the discretisation used as it has been stated in the manuscript. Based on these facts, WENO-5 should be the most accurate scheme as it is 5$^{th}$ order in space and 3$^{rd}$ order in time, but the high number of particles of MIC allow it to perform really well even though it is 2$^{rd}$ order in space, 1$^{st}$ order in time for the trajectory tracing and 1$^{st}$ order in space for the interpolation step in a similar way as an upwind scheme would perform really well if the resolution was high enough. In that light, what matters is not to fundamentally understand the order of accuracy of each scheme as they are based on fundamentally different approaches but how they perform for realistic geodynamical problems, which is the approach used in this work.

→ New simulations have been performed for a Courant number of 0.7 for testing the extended stability of the MIC and SL; results have been included in the revision.

I was confused by the fact that it was not possible to obtain MIC results using a 128 GB machine for grid resolutions such as 500x1000, when the number of markers is on the order of 10 million.

What was the exact reason for such limit in the current MIC implementation? The final conclusions build on the statement that the MIC scheme lacks mass conservation and it is expensive in terms of computation and memory. I already expressed my concerns about the observed lack of mass conservation in the case of the MIC scheme. I think that it would be also useful if the authors could elaborate more on the computational and memory performance of their MIC implementation.

Response: Unnecessary memory allocations were preventing the MIC to scale well.

→ These issues were solved and multithreading on CPU was implemented on all algorithms to further speed up the computation and allowing to perform all models at high resolution. The figures were updated in the draft with the new results.

Detailed comments:

l. 38 What is exactly the notion of triviality in this case?

Response: The meaning was that it is mathematically well understood.

→ This was clarified in the manuscript.

l. 43 "This brings limitation to the resolution of the model …" - It is indeed correct that for any given computational resources 1D numerical models can be studied with a higher resolution than 2D models. On the other hand, looking from a 3D physical perspective using approximate 2D numerical models could be considered as relaxing computational limitations to the resolution of the model.

Response: That is correct. We think that 2D models, in particular for two-phase flow in this context should be enough for most cases. But complex 2D models still have computational limitation, in particular if reactive transport wants to be achieved. In this case, it is still important to be able to implement accurate advection schemes.

eq.5 Is it necessary to use the brackets around $C_e^f$?

Response: → The brackets were removed.

l. 84 Perhaps early rather than earlier?

Response: → Corrected.

l. 87 "It consists at tracking individual particles on a Lagrangian frame and to reinterpolate them when needed on an Eulerian stationary mesh grid". consist at or consist of? to reinterpolate or reinterpolating?

Response: → Corrected to "consist of" and "reinterpolating".

l. 94 "..to the position of a fixed Eulerian grid.." It would be perhaps worth to mention grid nodes here.

Response: → Corrected.

l. 106 Please change spacial to spatial.

Response: → Corrected.

l. 106 It might be a bit confusing when the term "element" is mentioned here.

Response: → This has been replaced by chemical element

eq. 6 It would be perhaps more transparent if the f symbol, which stands for fluid, was used in the superscript rather than subscript. It could be mentioned that v denotes the x componentof the velocity.

Response: To stay consistent with all the other equations used for the schemes, we kept f as a subscript.

→ The explanation that it represents the x component of the velocity was added in the manuscript.

l. 109 Please consider using "grid spacing" rather than "increment in space".

Response: → This has been corrected.

l. 129 Here and in fig. 1 caption, "Upwind" is capitalized, while it is not in l. 111.

Response: → This has been corrected.

l. 129 Please consider removing the comma.

Response: → This has been corrected.

l. 133 Perhaps "for a single grid node" rather than "1 element" would suit better here. This issue reappears in other parts of the manuscript.

Response: → This has been clarified in the manuscript.

l. 153-54: What is meant by "complex problems" here? I suspect that for any problem boundary conditions need a special or careful treatment when a 5-point stencil is used. I don't think that it is necessary to present in details how BC are treated in this case, but a shortdescription would be useful.

Response: The authors agree that the mention of "complex" boundary conditions is confusing and misleading. Both Neumann and Dirichlet boundaries can be applied with WENO schemes, commonly done by using ghost points. The "complexity" mentioned refers to the fact that, as it is a 5 points stencil, 2 ghost points are required on each side of the model for boundaries other than periodic.

→ This has been clarified in the text and the confusing notion of complexity has been removed.

l. 158 It is not that important, but I'm wondering about the notion of "fully capture" in this case.

Response: This refers to the notion of "shock capture", which often arises from non-linear advection equations such as the Burger equation for example. It means not producing oscillations when dealing with shocks.

l. 164 Please replace spacial by spatial.

Response: → This has been corrected.

l. 171 What is "therefore" referring to?

Response: → This has been clarified in the text.

l. 182 " … is not easy to determine xd. A common approach to overcome this limit…" In my vewi, the term limit may be misleading in this context. I would suggest something like: A common approach to accurately determine …

Response: → This has been corrected.

l. 193 "a stencil point on the grid" Perhaps it could be described as "a grid node".

Response: → This has been replaced by "grid node".

l. 196 It could be several (neighboring) cells depending on the interpolation order (here cubic,so four nodes are needed to span the interpolation space). On the other hand, it depends onthe adopted definition of the cell. Anyway, I think that it would be worth to clarify this issue.

Response: → This was clarified in the manuscript.

l. 214 "…interpolate their values…". Perhaps "…interpolate field values associated with them…"

Response: → This has been corrected.

l. 220 "The initial value of each marker …" Perhaps "The initial field value in each marker …"

Response: → This has been corrected.

l. 222-23 The problem of deteriorating marker resolution is not only limited to highly divergent flows. It may also affect incompressible, but strongly stretching flows. In such cases, the density of markers can actually depend on the direction (markers may be locally jammed in acertain direction and rarefied in the direction that is perpendicular). Thus, simple scalar measures of marker density may fail in such cases.

Response: → The idea of this comment has been added to the manuscript.

L. 246-54 In my view, based on the presented description, it is not exactly clear how the field values are interpolated from markers to grid nodes. Is it the least-square type of linear interpolation? How many surrounding cells are used?

Response: The interpolation used is a a weighted distance averaging linear interpolant, similar as in Gerya, 2019 for instance using all the markers in the surrounding cells of a grid point i. We

agree that this was not detailed enough, especially that it is not a conventional interpolant.

→ A detailed explanation with the equations used was added in the manuscript.

l. 256-261 The domain size is given as dimensionless, while the time is in seconds.

Response: → This has been corrected.

l. 268 Isn't $S_k$ constant for all the internal cells?

Response: It is indeed.

→ $S_k$ was removed from the equation and also on equation 47.

l. 352-354 Is this the description of the ODE solver implemented in DifferentialEquations.jl (l. 348)? Please clarify.

Response: Yes, it is.

→ This has been clarified.

l. 363 The standard deviation is given as a dimensionless quantity, while the size of the model domain is given in meters.

Response: → This has been corrected; SD is also in metre.

l. 366-67 I might be missing something here, but I don't exactly understand how the initial distribution of the melt chemistry is governed. In fig. 8 it appears just bi-modal at t=0 Ma.

Response: The initial distribution of the melt chemistry is indeed bimodal. The initial andesitic composition is a circle of radius of 60 meter.

→ This has been added to the manuscript.

Table 3 Please fix the value of the shear modulus (an incorrect exponent is used).

Response: → This has been corrected.

l. 374-383 This part might be a bit confusing. Although the link to eq. 5 is mentioned in l. 320,I think that it would be useful to clarify which physical fields are advected in this model. Please explain why the oxide content may not be conserved, despite the melt fraction is claimed to be conserved due to the chosen numerical scheme. If the total mass of the melt composition (summed weight percentages?) is renormalized in each time step, then I'm not sure if it could be treated as a conserved quantity.

Response: The melt fraction is equal to the melt filled porosity ($\phi$) in the model, which is solved using Eq. 42 and Eq. 43 using finite differences and TR-BDF2 for the integration in time. This discretization is mass conservative as it is using conservative finite differences. So, the melt fraction is changing through time but not because of eq.5. What is advected by eq. 5 is the mass fraction of oxides $C_e^f$ in wt%. As we are using a non-conservative equation for the advection,

this mass fraction will not be conserved through time. As the conserved quantity is:

$$\phi \times C_e^f,$$

which represents the mass conservation of the chemical element in the melt (see eq. 4), and we know that the melt fraction is conserved (this was measured in the models), then only the conservation of wt% of the chemical element will be monitored.

The renormalization at each step may indeed prevent it from being considered as a conserved quantity from a theoretical point of view. To test that, the models were also run without the normalisation to see how this impact the mass balance. The results show that this effect is negligeable at the modelling scale for all the algorithms (at the order of the 9 decimals for all elements).

→ The part about which field is advected has been clarified in the manuscript.

l. 389 What is meant by mixture here?

Response: This term is incorrect.

→ Mixture will be replaced by "mixing".

l. 397 I am a bit confused by the observed lack of mass conservation in the upwind scheme. Is it due to the fact that the renormalization procedure is used? I would also expect MIC to be essentially mass conserving and non-diffusive. Are the observed mass loss effects due to marker reseeding and removal?

Response: The upwind discretization is not mass conservative because we solve a non-conservative form of the advection equation (eq. 5). It will be mass conservative only for simple velocity field and divergence-free. This is shown by the new test added to the revised manuscript. Eulerian methods can only be truly conservative if used on a flux form. The loss of mass of MIC is indeed due to the reseeding and removal, which is not mass conservative. An error was present in the implementation of the scheme, increasing the mass loss.

→ The mention that the upwind discretisation is not mass conservative for divergent flow has been added in the method part, and the MIC implementation error has been corrected in the revision. Here is the new figure obtained:

[Figure]

**Figure 10.** Mass conservation of silica content in the melt fraction for four different advection schemes and five different spatial resolutions at the end of each simulation. The Courant number used is 0.7 for WENO-5 and upwind and 0.7 or 1.5 for QMSL and MIC. The resolutions are 100×200, 200×400, 300×600, 400×800 and 500×1000. The physical parameters used are reported in Table 3.

l. 407 Using mid-point for computing trajectories in MIC and SL results in some iterations.

Response: → This has been corrected.

l. 423 What is exactly meant by complex boundary conditions?

Response: See previous comment on l.153-154.

→ This part will be reformulated in the manuscript.

Fig. 9 & 10 The axes are logarithmic and there is no need to add log(.) in the labels.

Response: → This has been corrected.

References:

Gerya, T. (2019). *Introduction to numerical geodynamic modelling*. Cambridge University Press.

---

## Author Comment (AC2)

03/05/2024 13:39:00"Modelling chemical advection during magma ascent" by Dominguez et al. is an interesting manuscript that compares four different numerical schemes for phase field advection, emphasizing their suitability within the context of two-phase flow models. I believe the manuscript is valuable for the earth sciences modeling community, as it assesses the behaviour of these advection schemes in terms of accuracy -here mass conservation- and computational cost.

The upwind and SL schemes are quickly disregarded as they introduce excessive numerical diffusion and have a low accuracy. The authors find WENO as the most suitable advection scheme as it conserves mass at a decent computational cost, arguing that MIC does not conserve mass efficiently well in one of their two numerical experiments and is far too expensive in terms of performance and memory consumption.

Below are some general comments regarding the manuscript, followed by detailed line by line comments.

Albert de Montserrat

**General comments:**

**(1**) The authors should provide more details about the interpolants used in both the Semi-Lagrangian and MIC schemes. They also mention that complex boundary conditions may require some extra care in the WENO scheme. It is not clear to me what these complex boundary conditions are. Do commonly used boundary conditions such as free-slip or no-slip require some special treatment? Since the periodic boundary conditions used in the examples given in the manuscript are not the only common ones, a more detailed discussion would be helpful.

Response: Bicubic B-spline was used for the semi-Lagrangian scheme and a weighted-distance averaging bilinear interpolant was used for MIC.

→ Details about them were added in the manuscript, in particular about MIC as it is a less conventional interpolant.

We agree that the mention of "complex" boundary conditions is confusing and misleading. Common boundary conditions, such as Neumann and Dirichlet boundaries can be applied with WENO schemes, commonly done by using ghost points. The "complexity" mentioned refers to the fact that, as it is a 5 points stencil, 2 ghost points are required on each side of the model for boundaries other than periodic.

→ This has been clarified in the text and the confusing notion of complexity has been removed.

**(2)** The rotating circle benchmark shows that WENO and MIC produce similar results in terms of mass conservation. However, WENO performs better in conserving mass in the two-phase flow example. It's important to note that in the first benchmark, WENO SL, and MIC are run with the same constant time step. In the other example, the Courant number for MIC was more than twice that of WENO's. To ensure a fair comparison of how well these schemes conserve mass, I suggest running the two-phase flow example with the same time step for

WENO, MIC and SL. This will make the comparison of mass conservation fairer. If the mass conservation of MIC and SL improves by using the same time step as WENO, it would be helpful to run both MIC and SL with increasing time steps to determine when their accuracy starts to deteriorate.

Response: → Following this suggestion, both models with SL and MIC were performed using a Courant number of 0.7 and 1.5. In addition, 2 implementation mistakes were found concerning the QMSL and MIC algorithms. This improved the general mass conservation of the algorithms. This will be added in the manuscript. However, the main conclusions of the paper remain unchanged.

**(3)** The manuscript concludes that, besides better mass conservation, the WENO scheme is the best candidate because of the poor performance of MIC and its excessive memory consumption. However, the comment stating that the MIC does not run at high resolution because it does not fit in 128 GB of RAM is surprising and unclear. Let's assume that one allows a maximum of 30 particles per cell, then in the higher resolution case you have an upper bound of 15e6 particles (500 x 1000 x 30). Further assuming that you are using FP64, this means that only 120MBs (in Julia: sizeof(Float64)*15e6 / 1e6 ) are needed to store a particle field. With this in mind, there is plenty of RAM available to store multiple fields in the particles, and even for a much larger number of particles. Last, all the necessary arrays can be pre-allocated so that all the MIC kernels (advection, interpolation, injection….) can be performed in-place without requiring additional memory

Inspecting your MIC algorithm with packages such as Cthulhu.jl and JET.jl will reveal type instabilities in the code, which result in dynamic dispatch and potentially generate a large number of unnecessary allocations. Available profiling tools can be used to track down where the remaining allocations come from.

The main point of this paper is not addressing the optimization of the implementation of any of the advection schemes here present. However, since performance is one of the criterions chosen by the authors to select the best candidate scheme, I would suggest that the authors consider resolving the main performance issues in their codes. By doing so, the MIC's performance is also likely to improve, enabling it to run the two-phase flow example at higher resolutions. This would allow for a better comparison between MIC and WENO, and potentially turn MIC into a more viable candidate.

Response: → following this comment, the main problem of memory in MIC was tackled and multithreading on CPU was implemented on all algorithms to further speed up the computation and allowing to perform all models at high resolution. The main slow down on MIC is now the injection and removal of markers that occur at almost every timestep in the two-phase flow case due to the highly divergent melt velocity field. The authors did not manage to fully parallelise this part as it requires to change the size of current vectors holding marker properties at run time (either to remove or to add new markers) as the high number of marker accumulating doesn't allow to preallocate them in a heuristic way.

**(4)** Attached is a PDF with a series of typo, grammar and style corrections and suggestions.

**Line by line comments**

L5: "weighted essentially non-oscillatory". Throughout the manuscript WENO is spelled out with either lower or upper case initials (e.g. line 125). For consistency, I recommend to use either one or the the other throughout the entire manuscript

Response: → This has been corrected.

L87 "It consists at" -> "It consists of"

Response: → This has been corrected.

L88 "to interpolate" -> "interpolating"

Response: → This has been corrected.

L88 "mesh grid". I would call it either mesh or grid.

Response: → The word "grid" is now used consistently.

L89 "to be" -> "being"

Response: → This has been corrected.

L133 "element" what is the element here?

Response: It is here the chemical element.

→ It has been clarified in the manuscript.

L147 You could consider using the machine precision epsilon (function eps() in Julia) instead of fixing epsilon=1e-6

Response: → This will be added in the manuscript.

L153 "Careful consideration must then be given to boundary conditions for complex problems." What kind of considerations have to be made for the boundary conditions? It would be helpful to elaborate on this. I don't think it is obvious.

Response: → This has been reformulated, as described in response to (1).

L166, Isn't the time integration third order in space and first order in time? If I'm reading correctly eqs. 11-13, C is evaluated at the same time step in all the integration stages.

Response: A time integration will only impact the order in time. You can see $C^2$ and $C^1$ as being intermediate timesteps with the coefficients in front of them and hence being third order in time. It should be seen as a modified 3rd order Runge Kutta scheme. For more detail, Gottlieb et al., 2001 provide the original derivation (eq 4.2 for this scheme in the original paper).

L176 "rectilinear grid" perhaps "uniform" or "regular" are more appropriate

Response: → modified to "regular grid".

L176 "this reduces the complexity of the implementation and the numerical cost of the interpolation function." Could you briefly describe why this is so (finding the parent cell, cheap mapping to the reference cell, etc…) ? It may not be obvious for a reader who never tried to implement any of these advection schemes.

Response: → An explanation will be added.

L177 "From a particle point of view, the goal is to find the starting points at the previous timestep for each grid point." Perhaps this could benefit from some rephrasing, what you want to do is to find where a particle that is at a grid point (i,j) at time $t^{n+1}$ was at the time $t^n$

Response: → This has been rephrased in the revised manuscript.

L183 What about higher order methods such as Runge-Kutta 4 ? would they work better to back track the particles?

Response: To our knowledge, mainly higher order linear multistep methods have been explored. For example, recently by Filbet and Prouveur, 2016 for backwards SL. They showed that using Adams-Moulton or Adams-Bashforth schemes improves the mass conservation and allow to take larger time steps when the resolution is big enough. This approach has the downside of also requiring values for the velocity at $t^{n-1}$. Runge-Kutta methods are not popular for this problem and the authors believe it is essentially for historical reasons, as SL schemes were originally developed for modelling the atmosphere (see Pursue and Leslie 1995 for more details on the advantages of multistep methods for this purpose). This could be interesting to explore as it may impact the conservation of mass of the scheme but this it is beyond the scope of this paper.

→ A more concise version of this answer was added in the discussion of the manuscript.

L190 How is vf interpolated to $t^{n+\frac{1}{2}}$? is it just the average between $t^n$ and $t^{n+1}$? I guess this requires storing the velocity at two time-steps, this should be mentioned somewhere in the manuscript.

Response: It can be either the average of $t^n$ and $t^{n+1}$ or extrapolated from $t^{n-1}$ and $t^n$ if the value at $t^{n+1}$ is not accessible. In this study, the mean is used as the momentum equations are solved prior to the advection equation.

→ A mention of this work has been added to the manuscript.

L197 Are bi/tri-linear interpolations not good enough?

Response: Using linear interpolation is equivalent to using an upwind scheme. Actually, for a Courant number lower than 1 and for a constant velocity field, it can be shown than upwind and linear SL schemes are equivalent (see Brasseur and Jacob, 2017 p.331 for instance).

L205 "clipping", I think "clamping" is a better word for this.

Response: Clipping is the common term used in the literature for this scheme. It is also consistent with the idea of limiting the maximum value.

L205-210 The manuscript lacks the definition of $C^H$. It would be helpful to include this information.

Response: As mentioned on line 205, $C^H$ is a high order interpolant. The definition is left as general as possible so it can be a cubic spline, or a fifth order hermite interpolation for example.

→ It has been clarified that it corresponds to a cubic spline in this study.

L214 "interpolate their values on a fixed grid." Interpolation often goes in both directions (e.g. temperature), not only from the markers onto the grid.

Response: → This is now mentioned in the manuscript.

L220 Are the initial positions of the markers randomized? If they are, it should be mentioned later on the benchmarks descriptions.

Response: They are not.

L220 "The initial value of each marker can then be obtained by linear interpolation from the initial conditions of the Eulerian grid." Most of the time, a field can be initialized directly at the particles, which provides a more accurate starting point

Response: → Correct, this will be clarified in the manuscript.

L223 "a **non-conservative** strategy"

Response: → This has been added.

L246-147 "This is more complex than for SL schemes because the markers are not uniformly distributed for a non-trivial velocity field." Another disadvantage is that the parallelizing this interpolation is susceptible to race conditions in shared memory systems, requiring the use of atomic operations.

Response: → This comment was added in the manuscript.

L252-253 Could you add eq. of the linear interpolant?

Response: → This will be added.

L263 How is the computational time of one time step measured? Is it the time of the first step? If so, does it include compilation time? Or is it the average step time throughout the whole model?

Response: The computation time is measured by running each scheme with the initial conditions 10000 times. The reported time is the minimum elapsed time measured (using @btime from Benchmark.jl) so it does not include the compilation time.

→ This procedure has been clarified in the manuscript.

As the number of particles varies throughout the model run time, it may be better to average the time of each time step (excluding the first one to avoid measuring compilation time, or doing a warm up run) and add a confidence interval.

Response: There is no reseeding or removal in the test as the number of particles doesn't go below the threshold of 25% of the original number of markers per cell and don't accumulate. We think that repeating the MIC with the initial conditions 10000 times is sufficient to capture the representative computational time of the MIC.

Would be helpful to clarify what is actually done and measured in the MIC case. I believe that you are interpolating from particles to grid (plus reseeding?) in each time step. This is technically not needed in this case, since you are merely advecting passive markers. It may be helpful to include in the manuscript a breakdown of the percentage of time spent on all the MIC stages done in the benchmark.

Response: The measured time include the interpolation from particles to grid.

Table 2.

- Here you call it "SL QM", however, it's written as "QM SL" in some other places. Please choose one naming convention for the whole manuscript.

  Response: → This has been corrected.

- I get very different timings for SL and MIC (Julia 1.10, Windows 11, CPU: 24 × AMD Ryzen 9 5900X 12-Core Processor). Different Julia versions are known to have different performances, it would be helpful to add to the caption what Julia version was used. Later on you mention what CPU you used, but I think it would be nice to have this information also in this caption.

  Response: The timings were corrected, with single and multithreaded cases added. → The Julia version (1.10.2) and the CPU details were also added to the caption.

Section 4.2 Perhaps it is better to define the scaling lengths in a small table instead of spelling them out line by line.

Response: → The scaling variables has been added to Table 3 and removed from the text.

Section 4.3 The time step of QMSL and MIC is roughly ~x2 with respect to WENO and Upwind. It may be beneficial to run QMSL and MIC with the same Courant number as in the other two cases and include the mass conservation results to Fig 9.. This could potentially reduce the reseeding in the MIC case and improve the mass conservation.

Response: → Following this concern, both QMSL and MIC were also performed with a Courant number of 0.7. Results will be included on Fig 10 and 11.

L349 "This method" -> "This package"

Response: → This will be corrected.

L354 As a curiosity, what kind of automatic differentiation is being used?

Response: Forward automatic differentiation using ForwardDiff.jl.

→ This was added to the manuscript.

L360 "assert" means to confirm something. Perhaps "examine", "test", or similar, work better here.

Response: → It will be replaced by assess

L365 It is stated later that all the runs are running serially in a single core, perhaps it's useful to mention that fact here instead/as well.

Response: → The models are now performed using multithreading. This will be mentioned there.

L399 "that is linked on how often the two-phase flow solver is called" perhaps you could rephrase it so that it is more clear that being able to increase the time step results in fewer time steps being required.

Response: → This has been clarified.

L400 "...QMSL shows the best scaling…" What do you mean by scaling here? Scaling is usually referred to as how well one algorithm scales with the number of cores/CPUs. However, all the models were run serially in a single core.

Response: The scaling referred here to the increase in resolution as QMSL was the fastest algorithm at high resolution due to the high Courant number used.

→ The word scaling was removed to clarify the meaning.

L423 "If complex boundary conditions are required" As in my previous comment, I think it is not obvious what these complex boundary conditions are.

Response: → This has been reformulated, as described in response to (1).

References:

Gottlieb, S., Shu, C. W., & Tadmor, E. (2001). Strong stability-preserving high-order time discretization methods. *SIAM review*, *43*(1), 89-112.

Filbet, F., & Prouveur, C. (2016). High order time discretization for backward semi-Lagrangian methods. *Journal of Computational and Applied Mathematics*, *303*, 171-188.

Brasseur, G. P., & Jacob, D. J. (2017). *Modeling of atmospheric chemistry*. Cambridge University Press.

Purser R. J. & Leslie L. M. (1995). Accuracy and conservation in semi-Lagrangian time-integration. *ECMWF*

---

## Author Response (AR2)

Dear editor,

The conclusion was modified accordingly to the comment of one of the reviewers to reflect more why the authors believe the WENO-5 scheme is the best scheme for this study with respect to the MIC and QMSL schemes. It was done by highlighting the high number of reseeding and removal of markers needed by using the MIC scheme, impacting the performance of this scheme, and the fact that the QMSL shows the worst mass conservation between the three schemes.

Hugo Dominguez on behalf of the authors